# Online Surface Roughness Prediction for Assembly Interfaces of Vertical Tail Integrating Tool Wear under Variable Cutting Parameters

**DOI:** 10.3390/s22051991

**Published:** 2022-03-03

**Authors:** Yahui Wang, Yiwei Wang, Lianyu Zheng, Jian Zhou

**Affiliations:** School of Mechanical Engineering and Automation, Beihang University, Beijing 100191, China; wangyahui@buaa.edu.cn (Y.W.); wangyiwei@buaa.edu.cn (Y.W.); zhouj@buaa.edu.cn (J.Z.)

**Keywords:** assembly interfaces, surface roughness prediction, varying tool wear, SAE–LSTM, transfer learning, variable cutting parameters

## Abstract

Monitoring surface quality during machining has considerable practical significance for the performance of high-value products, particularly for their assembly interfaces. Surface roughness is the most important metric of surface quality. Currently, the research on online surface roughness prediction has several limitations. The effect of tool wear variation on surface roughness is seldom considered in machining. In addition, the deterioration trend of surface roughness and tool wear differs under variable cutting parameters. The prediction models trained under one set of cutting parameters fail when cutting parameters change. Accordingly, to timely monitor the surface quality of assembly interfaces of high-value products, this paper proposes a surface roughness prediction method that considers the tool wear variation under variable cutting parameters. In this method, a stacked autoencoder and long short-term memory network (SAE–LSTM) is designed as the fundamental surface roughness prediction model using tool wear conditions and sensor signals as inputs. The transfer learning strategy is applied to the SAE–LSTM such that the surface roughness online prediction under variable cutting parameters can be realized. Machining experiments for the assembly interface (using Ti6Al4V as material) of an aircraft’s vertical tail are conducted, and monitoring data are used to validate the proposed method. Ablation studies are implemented to evaluate the key modules of the proposed model. The experimental results show that the proposed method outperforms other models and is capable of tracking the true surface roughness with time. Specifically, the minimum values of the root mean square error and mean absolute percentage error of the prediction results after transfer learning are 0.027 μm and 1.56%, respectively.

## 1. Introduction

Surface quality has a critical impact on the reliability and lifetime of high-value products [1,2,3], such as rockets, spacecraft, and aircraft. In particular, the surface quality of the assembly interfaces of these products directly affects the final product quality. For example, the assembly interfaces of an aircraft connect adjacent large-scale aircraft components (e.g., wings, tails, and fuselages). The assembly interfaces of an aircraft’s vertical tail are shown in Figure 1. The vertical tail assembly interfaces consist of eight sub-assembly interfaces. The machining process mainly includes milling the assembly interface plane and drilling the connecting holes. Assembly interfaces are characterized by a wide distribution distance, difficult-to-cut materials, and an uneven machining allowance distribution, leading to severe tool wear during the machining process. Consequently, surface quality is difficult to control and directly measure [4]. In aircraft assembly, ensuring the surface quality of assembly interfaces is critical to the final quality of the aircraft. There are many parameters affecting surface quality, such as surface roughness (2-D and 3-D surface roughness), surface waviness, surface form, etc. [5,6], among which, surface roughness parameters and the associated functionality are very important for the evaluation of surface integrity and machining quality [7,8] since surface roughness significantly influences the assembly accuracy, fatigue strength, corrosion resistance, and contact stiffness of the parts [9,10]. Consequently, surface roughness becomes an important parameter of concern for engineers during aviation manufacturing [11,12], for instance, in the manufacturing of the assembly interface of an aircraft vertical tail, as shown in Figure 1. However, the machining of assembly interfaces depends considerably on the experience of workers who assess the tool wear and surface quality as well as adjust the machining parameters. To optimize the machining process and implement the online adjustment of parameters, the online monitoring of machined surface roughness, Ra, has practical significance. Accordingly, the motivation of this study is to explore the online monitoring method of surface roughness, Ra, through cutting experiments to achieve the online monitoring of assembly interface machining.

Surface roughness is typically monitored either offline or online. The offline approach involves the use of contact measurement equipment [13,14] and optical measurement equipment [15,16,17]. The offline approach has drawbacks, such as long measurement time, specific requirements of the working environment, complexity to set up, etc. In contrast, online methods are not hindered by the aforementioned disadvantages. In light of this, they have attracted increasing research interest [18,19,20]. Online methods aim to establish a mapping model (prediction model) between surface roughness and online monitoring data, such as cutting parameters and sensor signals [21]. In addition, there are also studies on surface roughness estimation through the numerical method, which is difficult to integrate the real-time sensor data of the machining site to adaptively adjust to the roughness prediction online. Given the cons of offline and numerical methods, this paper uses a scheme for surface roughness online prediction based on real-time sensor data.

Currently, research on online monitoring has several deficiencies. In practical machining, surface roughness varies. The factors affecting surface roughness include the machine tool, workpiece, tool characteristics, dynamic parameters, etc., among them, the tool that is used to form workpiece surface in machining, which directly affects the surface roughness [22]. For example, as the tool wears, the contact angle between the tool and workpiece changes, which deteriorates the surface roughness of the workpiece [23]. However, the effect of tool wear on surface roughness has rarely been considered in current research [23,24]. In addition, the deterioration trend of surface roughness and tool wear differs with variable cutting parameters. Consequently, a prediction model trained with the monitoring data collected under one group of cutting parameters can fail to accurately predict roughness when the cutting parameter changes. Evidently, it is unrealistic to train prediction models with all possible cutting parameters. Therefore, to develop a practical surface roughness prediction model, the inclusion of tool wear as one of the variable cutting parameters is necessary. Note that previous work [25] introduced, in detail, a method and process for tool condition monitoring. This current study focuses on the surface roughness prediction method. In this paper, the formulation and training of the surface roughness prediction model and its prediction process integrating tool wear are described in detail.

With the above motivation, an online surface roughness prediction method considering tool wear under variable cutting parameters is proposed. In this method, first, a framework of surface roughness prediction for assembly interface is proposed. Subsequently, a surface roughness prediction model based on transfer learning is established. Finally, the model input considering tool wear and the prediction process of surface roughness are designed. This proposed method realizes the online prediction of surface roughness considering the influence of tool wear under variable cutting parameters, which is more accurate and practical than the existing methods using sensor data alone.

The remainder of the paper is organized as follows: Section 2 reviews related work. In Section 3, the framework of surface roughness prediction is proposed. Subsequently, this same section details the structure of the surface roughness prediction model, its training strategy based on transfer learning, and the integration of tool wear prediction into surface roughness prediction. The experimental settings and the performance of the proposed method under variable cutting parameters are presented in Section 4. The conclusion and future work are discussed in Section 5.

## 2. Related Work

In this section, first, surface roughness online prediction methods are reviewed as indirect and direct sensor-based techniques depending on whether or not surface roughness is directly measured. Next, the advantages and disadvantages of the current online surface roughness prediction methods are summarized. Finally, the problems solved in this paper and specific contributions are given. 

### 2.1. Surface Roughness Prediction with Indirect Sensors

This class of methods attempts to establish the mapping between the surface roughness and indirect sensor signals (e.g., vibration, cutting force, current, and sound pressure) monitored during the machining process. Commonly used prediction methods include polynomial regression models and artificial intelligence models.

#### 2.1.1. Polynomial Regression Model

Wang et al. established a polynomial regression model for surface roughness prediction considering tool geometry and cutting force as factors for Ti6A14V. The model has high applicability and accuracy for predicting milling surface roughness [26]. However, it only considers the combination of several groups of discrete cutting parameters and does not account for predicting surface roughness at different times during machining. The use of sensor data can reflect the surface roughness changes caused by time-varying factors in machining. To realize surface roughness prediction during the cutting process through a polynomial regression analysis, some researchers considered cutting force, vibration, and cutting energy consumption [27,28,29]. In the turning process of the alloy material Inconel 718, Deshpande used the polynomial regression analysis to combine cutting parameter data and sensor data (e.g., cutting force, vibration, etc.) in the cutting process to predict surface roughness [30].

The abovementioned research adopted regression models to accurately reflect the surface roughness trend under certain conditions. However, the studies only considered the sensor data during cutting and neglected the time-varying factors, such as tool wear, which will affect the accuracy of roughness prediction.

#### 2.1.2. Artificial Intelligence Model

The following studies on surface roughness prediction are based on the information obtained by a single sensor. Guo and Wu employed the specified time–frequency domain features of the vibration signal of the cutting process as input and then utilized the long short-term memory network (LSTM), convolutional neural network (CNN), and other neural network technologies to predict surface roughness [31,32]. Huang proposed an online modeling and surface roughness monitoring system based on gray theory and a bilateral best-fitting method by monitoring the cutting force signal in the cutting process [33]. Gerardo used a combination of incremental modeling and simulated annealing to model cutting forces and parameters to predict the surface roughness in milling machining [34]. In addition, particle swarm optimization–support vector machine [35], differential evolution algorithm [36], singular spectrum analysis, and related principal component analysis [37] are used to predict surface roughness based on the vibration signals in the cutting process.

The following studies on surface roughness prediction are based on multi-sensor information. Sun et al. designed a surface roughness prediction model that includes an embedded neural network and an output neural network, which is able to obtain superior prediction accuracy than a single neural network [38]. Kumar considered the cutting parameters and sensor signals as control variables and predicted surface roughness by response surface methodology and an artificial neural network. The results showed that the prediction accuracy of utilizing sensor signals is better than merely considering cutting parameters [39].

The aforementioned research on surface roughness prediction based on sensor data and intelligent algorithms has enabled online prediction. Although the variations in tool wear and cutting parameters directly influence the prediction of surface roughness, these are rarely considered by current research.

### 2.2. Surface Roughness Prediction with Direct Sensors

The direct-sensor-based surface roughness prediction method predicts roughness through the visual image of a machined surface.

The research method mainly involves predicting surface roughness according to the reconstructed surface topography or image feature of the machined surface. Liu proposed a new surface roughness measurement method based on a color distribution statistical matrix [40]. Prabhakar proposed a hybrid transform method that combines fast Fourier transform, discrete wavelet transform, and discrete shearlet transform to achieve surface roughness prediction [41]. Tootooni used the algebraic graph theory image processing method to convert surface images into unweighted, undirected network graphs and estimated the graph theory invariant Fiedler number (λ2) as a surface roughness discriminator [42]. Chiou proposed a surface texture model based on visual data for real-time, remote, automatic detection of surface quality [43]. Jeyapoovan used charge-coupled device cameras and multi-color light sources to capture the images of machined surfaces with different surface roughness values [44]. Shahabi proposed a method for surface roughness measurement using a two-dimensional profile extracted from an edge image of the workpiece surface [45].

Surface roughness prediction using direct sensors can directly obtain surface roughness through workpiece surface images. However, illumination, cutting fluid, and dust in the actual cutting environment can interfere with image processing, thereby affecting the prediction.

### 2.3. Summary and Analysis

In contrast with the prediction techniques based on the indirect use of sensors, direct sensor-based surface roughness prediction methods are considerably affected by illumination, cutting fluid, and dust. The proposed method belongs to the category of an artificial intelligence model, which can make full use of real-time sensor data to drive the model to predict surface roughness online. There are still several limitations in the current research. In variable cutting parameter machining, the deteriorating trend of tool wear with time is different and has a deteriorating effect on surface roughness. However, there is no mature solution for online surface roughness prediction that considers both tool wear and variable cutting parameters. To resolve these problems, this paper proposes a surface roughness prediction model that considers the tool wear variation with variable cutting parameters. The contributions of this study are as follows: (1) The time-varying characteristics of sensor data and tool wear are considered to predict the surface roughness of assembly interfaces, rendering surface roughness predictions more suitable for practical machining; (2) the multi-time step input stacked autoencoder (SAE) is designed to extract the time series features of raw input data and, combined with the LSTM, to predict the surface roughness degradation value in assembly interface machining; and (3) an SAE and LSTM (SAE–LSTM) surface roughness prediction model based on transfer learning is proposed that resolves the problem of surface roughness online prediction for vertical tail assembly interfaces by considering variable cutting parameters.

## 3. Proposed Surface Roughness Prediction Method Considering Tool Wear

The framework and working mechanism of the surface roughness prediction for assembly interfaces are presented in Section 3.1. Next, the structure of the fundamental roughness prediction model, SAE–LSTM, is described in Section 3.2. The training strategy based on transfer learning for resolving the problem of the variable cutting parameters is elaborated on in Section 3.3, and the integration of tool wear prediction into surface roughness prediction is discussed in Section 3.4.

### 3.1. Framework and Working Mechanism of Surface Roughness Prediction for the Assembly Interface

Before elaborating on the proposed framework of surface roughness prediction, some specific terminologies used are introduced, as summarized in Table 1. As shown in Figure 2, the framework has three parts: pre-training of the surface roughness prediction model in the source domain, transfer learning for the modules of the source domain model, and predicting the surface roughness in the target domain. The effects of surface roughness prediction are shown in Figure 2d. The prediction of surface roughness for the assembly interface considering variable cutting parameters can be achieved via transfer learning.

As shown in Figure 2a, a fundamental surface roughness prediction model (known as the source domain model) is first trained on the labeled source domain data. The input training data are sensor data, such as vibration and current signals, and the output data are ground truth labels (i.e., the surface roughness values of assembly interfaces measured offline by a contact device under the source domain). Subsequently, the basic surface roughness prediction model gains the ability to predict the surface roughness of the assembly interface in real time using the online monitoring data.

Next, the transfer learning strategy is applied to the modules of the source domain model, as shown in Figure 2b and detailed in Section 3.3. After applying transfer learning, a surface roughness prediction model (known as target domain model) is derived. This model is capable of predicting the surface roughness value of the assembly interface under the target domain.

The integration of tool wear prediction into surface roughness prediction under the target domain is shown in Figure 2c. The predicted tool wear value and collected sensor data are integrated to form the target domain data. Next, the target domain model obtained in Figure 2b is used to predict the surface roughness in the scenario that accounts for tool wear. The surface roughness and tool wear of the assembly interface show different, increasing trends under the source and target domains, proving the necessity of using transfer learning, as shown in Figure 2a,c.

The prediction results of the surface roughness of the assembly interface obtained with and without the use of transfer learning are shown in Figure 2d. The source domain model has a satisfactory prediction performance under the source domain; however, it fails to predict the surface roughness value under the target domain, exhibiting a considerable deviation from the true surface roughness values. After undergoing transfer learning, the surface roughness prediction model became capable of more accurately predicting the surface roughness value under the target domain.

### 3.2. Fundamental Surface Roughness Prediction Model Based on SAE–LSTM

Before establishing the model, it is necessary to discuss the consideration of the model input data in this paper. The research shows that in machining dynamics, tool wear has an impact on the surface morphology generation [46], cutting force [47], contact stress, chatter, temperature [48,49,50], etc. Therefore, it is essential to comprehensively consider the sensor data that reflect the actual machining situation and the tool wear data.

The fundamental surface roughness prediction model is based on SAE–LSTM; its structure is shown in Figure 3b. This prediction model includes four components: input data, a feature extraction module, a prediction module, and an output module.

The SAE is an unsupervised, deep learning network widely used for data dimensional reduction and feature extraction. Research indicates that it has satisfactory performance for extracting time series features [51]. The SAE structure is shown in Figure 3a. The “Encoder” consists of three convolutional layers (Conv) and three max-pooling layers (MP); the “Decoder” includes three convolutional layers and three upsampling layers (US). To extract the time series features of raw input data, the input of SAE is designed as a multi-time step input (input data are elaborated in Section 3.4.2), and the SAE structure is designed by a one-dimensional CNN. Note that the SAE is first trained and then only the “Encoder” of this trained SAE is used as the feature extraction module. In this way, the ability of SAE to extract features from the raw input data can be used to reduce the dimension of the data and improve the training efficiency of the model. In addition, due to the excellent time series prediction capability of the LSTM, the three-layer LSTM is employed to predict the surface roughness of the assembly interface in the prediction module, and the “Flatten” and “Repeat Vector” layers are utilized to adjust the data dimension to fit the LSTM layer. Finally, the dense layer is used to output the predicted value.

The training data of the SAE–LSTM are as follows. The input data are composed of four types of sensor data, i.e., tool wear (*VB*), sound pressure (*Pa*), vibration signal (*V*), and spindle current (*C*), which are acquired during machining. The surface roughness prediction model is trained with the time sequences of tool wear data and sensor data, i.e., ([${VB}_{t}^{S}$, ${Pa}_{t}^{S}$, $V_{t}^{S}$, $C_{t}^{S}$], …, [${VB}_{t+n}^{S}$, ${Pa}_{t+n}^{S}$, $V_{t+n}^{S}$, $C_{t+n}^{S}$]), where ${VB}_{t}^{S}$ is the tool wear value at time *t* in the source domain; ${Pa}_{t}^{S}$, $V_{t}^{S}$, and $C_{t}^{S}$ denote the sound pressure, vibration, and spindle current signals, respectively. The surface roughness predicted at different times under the source domain is denoted as [${\hat{Ra}}_{t}^{S},{\hat{Ra}}_{t+1}^{S},\ldots ,{\hat{Ra}}_{t+n}^{S}$].

### 3.3. Surface Roughness Prediction Model Training Using Transfer Learning

#### 3.3.1. Transfer Learning Strategy Based on a Multi-Stage Model Training Process

To realize surface roughness prediction considering variable cutting parameters, a transfer learning strategy based on a four-stage model training process is proposed. Through transfer learning, the model trained with source domain data can be easily used to predict surface roughness with the target domain data, thus improving the application scope of the surface roughness prediction model. As shown in Figure 4, the first two stages are pre-training stages, and the latter two are fine-tuning stages.

In the first stage, the SAE is trained with a part of the source domain data such that it gains the ability to extract features from the source domain data. In the second stage, the weight of the “Encoder” of the trained SAE is loaded; it serves as the feature extraction module of the SAE–LSTM. Next, a part of the source domain data and corresponding labels are used as the input–output pairs to pre-train the SAE–LSTM such that the source domain model (pre-trained SAE–LSTM) gains the ability to predict the surface roughness in the source domain.

In the fine-tuning stage, this paper needs to fine-tune the feature extraction module and prediction module based on the source domain model. In the third stage, a double-input structure model based on the source domain model is designed. The source and target domain data are used as inputs of the double-input model. The feature extraction module extracts features from the two domains. It should be noted that the target domain data here does not need label data. The distribution difference of features extracted from the two domains is computed in a customized layer, which quantifies the distribution difference between the source and target domain data using the maximum mean discrepancy (MMD). Next, the error between the distribution difference and zero is back-propagated to optimize the parameters of the SAE–LSTM model. Accordingly, the source domain model can adapt to the target domain. However, due to the huge difference between the source domain and the target domain, only using the unlabeled target domain data to train the feature extraction module cannot obtain a high-precision prediction effect. The strategy of using a small number of target domain label data to adjust the prediction module has been proven to improve the efficiency and accuracy of model training in the transfer learning method [52], and a small amount of label data in the target domain is usually easy to obtain. Therefore, in the fourth stage, the model is adjusted to a single input, and the weight of the fine-tuned feature extraction module is frozen so that the prediction module only participates in the fine-tuning. In this way, it can obtain high prediction accuracy without retraining the model with a large number of label data in the target domain.

#### 3.3.2. Optimization Objective of Multi-Stage Transfer Learning

The pre-training and fine-tuning processes include two loss functions, L1 and L2. These two loss functions are based on the mean absolute error (MAE). When the model has a single-input structure, such as that shown in Figure 5 and Figure 6b, the loss is L1; when the model has a double-input structure (Figure 6a), the loss is L1 + L2.

The SAE and SAE–LSTM are the basis of the surface roughness prediction model, and their precision resulting from the two-domain data training determines the final prediction accuracy. The loss function, L1, is used to train the SAE and the single-input SAE–LSTM, as shown in Equation (1), where *n* is the number of training samples, and $y_{i}$ and ${\hat{y}}_{i}$ are the ground truth and predicted values of surface roughness, respectively.
(1)$$L_{1}=\frac{1}{n}\sum_{i=1}^{n}\left|y_{i}-{\hat{y}}_{i}\right|$$

The role of L2 is to reduce the distribution difference between the source and target domain data through the maximum mean discrepancy (MMD). The MMD was first proposed by Gretton et al. to test whether the two distributions, *p* and *q*, differ by extracting samples from each of them [53]. In terms of transfer learning, the MMD is used as a metric to measure the difference between the source and target domain data. Under different machining conditions (e.g., variable cutting parameters), the source domain data (*X^S^*) and target domain data (*X^t^*) are assumed to be two random variables following two different distributions; $x_{i}^{s}$ and $x_{i}^{t}$ are observations of *X^S^* and *X^t^*, respectively.

The square of the MMD between the two distributions, denoted as ${\hat{D}}^{2}$, is computed by Equation (2), where the Gaussian kernel function is used; σ is the kernel bandwidth; *N* is the number of samples.
(2)$$\begin{array}{c} {\hat{D}}^{2}[X^{s},X^{t}]=\frac{1}{N\left(N-1\right)}\sum_{i\neq j}^{N}\exp(-||x_{i}^{s}-x_{j}^{s}|{|}^{2}/2\sigma^{2})\\ -\frac{2}{N^{2}}\sum_{i=1}^{N}\sum_{j=1}^{N}\exp(-||x_{i}^{s}-x_{j}^{t}|{|}^{2}/2\sigma^{2})\\ +\frac{1}{N\left(N-1\right)}\sum_{i\neq j}^{N}\exp(-||x_{i}^{t}-x_{j}^{t}|{|}^{2}/2\sigma^{2}) \end{array}$$

Loss L2 is obtained by calculating the MAE between ${\hat{D}}^{2}$ and the zero-valued label.
(3)$$L_{2}=\frac{1}{n}\sum_{i=1}^{n}\left|{\hat{D}}_{i}^{2}-0\right|=\left|{\hat{D}}^{2}[X^{s},X^{t}]-0\right|={\hat{D}}^{2}[X^{s},X^{t}]$$

The multi-stage transfer learning is detailed below. The schematic of each stage is given, as shown in Figure 5 and Figure 6. The schematic of SAE pre-training is shown in Figure 5. The source domain data are used as the SAE input. The purpose is to train the SAE by utilizing source domain data such that the encoder gains the ability to extract features from the input data. The SAE is optimized by minimizing loss L1 in which “Os” and “Is” are the output and input of SAE in the source domain, respectively (recall that the SAE is an unsupervised model). The schematic of SAE–LSTM pre-training is shown in Figure 5b; the source domain data are utilized to train this surface roughness prediction model. The SAE–LSTM is optimized by minimizing the loss, L1, in which “Os” and “Ls” are the outputs of SAE–LSTM in the source domain and ground truth (i.e., the label data of source domain), respectively.

After the pre-training process shown in Figure 5, the SAE–LSTM becomes capable of predicting the surface roughness of the source domain. As shown in Figure 6, the source domain model is transferred to predict the surface roughness of the target domain by fine-tuning the feature extraction and prediction modules.

To fine-tune the feature extraction module, the SAE–LSTM is first redesigned as a double-input model, and the weights of the source domain model are loaded. Second, the SAE–LSTM prediction module is frozen, the source and target domain data are used as double inputs, and the source domain label data and zero-value label are used as double outputs. The double-input SAE–LSTM is optimized by minimizing the loss L1 + L2. The output of SAE–LSTM in the source domain and the source domain label data are denoted as “Os” and “Ls,” respectively; “Lz” is the zero-value label. After fine-tuning the feature extraction module, the feature extraction capabilities of the SAE–LSTM in the source domain data are transferred to the target domain data.

However, due to the considerable difference in surface roughness under different cutting parameters, the SAE–LSTM prediction module still carries the risk of prediction error. Therefore, as shown in Figure 6b, an extremely small amount of target domain label data is used to fine-tune the prediction module, thereby improving the predictive ability of the SAE–LSTM in the target domain. The SAE–LSTM is optimized by minimizing loss L1, in which the SAE–LSTM output under the target domain and the target domain label data are represented by “Ot” and “Lt,” respectively.

### 3.4. Integration of Tool Wear Prediction into Surface Roughness Prediction

#### 3.4.1. Surface Roughness Prediction Integrating Tool Wear

As presented, considering the tool wear in surface roughness prediction is necessary. Two consecutive steps are involved. First, the time series of tool wear values and sensor data prior to timestamp *t* is used to predict the tool wear value at time *t*, ${\hat{VB}}_{t}$. Subsequently, the time series of sensor data, along with ${\hat{VB}}_{t}$, is used to predict the surface roughness at time *t*, ${\hat{Ra}}_{t}$; details are provided by Equations (4)–(7). The mapping between the input and output for tool wear prediction is given by Equation (4), where *F_VB_* denotes the tool wear prediction model, $X\_VB_{t-k},X\_VB_{t-k+1},\ldots ,X\_VB_{t-1}$ are the input time series, and *k* is an important parameter representing the length of historical data prior to the current time that is used as the model input to predict the future. The input, *X_Ra*, is a vector containing 11 physical quantities, as given in Equation (7), including the tool wear prediction value ($\hat{VB}$) and sensor data of ten channels: sound pressure (*Pa*) signal, vibration signals in three-axis on the spindle and spindle box (*V1x*, *V1y*, *V1z*, *V2x*, *V2y*, and *V2z*), and the three-phase current signal of the spindle controller (*Cu, Cv,* and *Cw*). The unit of tool wear (*VB*) and surface roughness (*Ra*) is microns (μm), the unit of the sound pressure signal is Pascal (Pa), the unit of vibration signal is gravity acceleration 9.8 m/s^2^ (g), and the unit of the current signal is Ampere (A). The values of *VB* and *Ra* are actual measured or predicted values, and the measured values of the sensor signal are given in Section 4.1. The mapping between the input and output for surface roughness prediction is given by Equation (5), where ${\hat{Ra}}_{t},{\hat{Ra}}_{t+1},\ldots ,{\hat{Ra}}_{t+m-1}$ represents the predicted surface roughness value sequence at time *t*; *m* represents the number of predicted surface roughness values.
(4)$$\left({\hat{VB}}_{t},{\hat{VB}}_{t+1},\ldots ,{\hat{VB}}_{t+m-1}\right)=F_{VB}\left(X\_VB_{t-k},X\_VB_{t-k+1},\ldots ,X\_VB_{t-1}\right)$$
(5)$$\left({\hat{Ra}}_{t},{\hat{Ra}}_{t+1},\ldots ,{\hat{Ra}}_{t+m-1}\right)=F_{Ra}\left(X\_Ra_{t-k+1},X\_Ra_{t-k+2},\ldots ,X\_Ra_{t}\right)$$
(6)X_VB=[Pa,V1x,V1y,V1z,V2x,V2y,V2z,Cu,Cv,Cw]
(7)$$X\_Ra=\left[\hat{VB},Pa,V1x,V1y,V1z,V2x,V2y,V2z,Cu,Cv,Cw\right]$$

The two consecutive steps mentioned above, considering *k* = 1 and *m* = 1, for instance, are illustrated in Figure 7. At time *t*, the tool wear value of target domain ${\hat{VB}}_{t}^{T}$ is predicted by the tool wear prediction model, *F_VB_*, using the data acquired at *t*−1. Next, ${\hat{VB}}_{t}^{T},\;$ along with the sensor signal collected at time *t*, is used as the input of the surface roughness prediction model, *F_Ra_*, to predict the surface roughness at time *t*, ${\hat{Ra}}_{t}^{T}$. The above process iterates with time such that the surface roughness integrating tool wear is predicted. The effect of the surface roughness prediction scheme is evaluated by the root mean square error (*RMSE*) and mean absolute percentage error (*MAPE*) calculated by Equations (8) and (9), respectively; *n* is the number of samples, $y_{t}$ is the surface roughness value measured offline by a contact instrument (which is regarded as the ground truth), and ${\hat{y}}_{t}$ is the predicted surface roughness value.
(8)$$RMSE=\sqrt{\frac{\sum_{t=1}^{n}{\left({\hat{y}}_{t}-y_{t}\right)}^{2}}{n}}$$
(9)$$MAPE=\frac{1}{n}\sum_{t=1}^{n}\left|\frac{{\hat{y}}_{t}-y_{t}}{y_{t}}\right|\times 100\%$$

#### 3.4.2. Data Preparation for Surface Roughness Prediction

The data preparation of the two consecutive steps and surface roughness prediction process in the target domain are shown in Figure 8. In Figure 8a, *k* = 5 and *m* = 1. For the input, *X_VB*, of the tool wear prediction model, the standard deviation and root mean square (RMS) features extracted from raw signals are found to have satisfactory monotonicity and can well predict the tool wear based on previous research [25]. The tool wear (*VB*), surface roughness (*Ra*), and the RMS of sensor data are shown in Figure 9. The monotonicity of most feature data is similar to *VB*; therefore, the two features above are selected for the sensor data features (*Pa, V*, and *C*). The data format of *X_VB* is shown in Figure 8a; the data dimension is (5,20).

The data generation process for surface roughness prediction integrating tool wear in the target domain is shown in Figure 8b; the surface roughness prediction model is given by Equation (5). The time lap steps are *k* = 5 and *m* = 1. The raw sensor data points obtained at 1-s intervals are selected as input data of the surface roughness prediction model. In addition, the tool wear prediction value is copied in sensor data form to combine it with the sensor data. As shown in Figure 8b, the input data dimension of the target domain is (50,000,11) based on the sampling frequency (Section 4.1).

## 4. Experiment and Analysis

### 4.1. Experimental Setup

The vertical tail of the aircraft and its assembly interfaces are shown in Figure 1 and Figure 10a. The assembly interfaces consist of eight sub-assembly interfaces; the material is Ti6Al4V. The finish machining operation for assembly interfaces is typically implemented to guarantee the final assembly quality. According to previous studies and the experience of workers in machining the assembly interfaces of the vertical tail, the typical machining parameters under dry cutting conditions are as follows: spindle speed, 150–500 r/min; axial cutting depth, 0.1–0.7 mm; and feed rate, 90–250 mm/min [54,55]. Due to the strict management of the machining process of the aircraft assembly interface, it is difficult to collect sensor data during the machining process of the real assembly interface. However, the material of the assembly interface, machining parameter range, tool type, surface quality requirements, geometric dimensions, and other information of the vertical tail assembly interface are available. This information is enough for us to design and fabricate a highly similar sample workpiece of the assembly interface, simulate the machining process of the real assembly interface in the experimental environment, and design a complex experimental data acquisition scheme. Therefore, the milling experiment of the sample workpiece is adequately capable of simulating the real assembly interface, and the proposed method can be validated by collecting the data during machining.

It should be pointed out that the research problem of this paper comes from the machining of a certain type of vertical tail assembly interface, which is made of Ti6Al4V. In addition, the machining experiment process is costly in terms of complexity and time. Therefore, this paper uses the material Ti6Al4V to carry out the machining experiments and verify the proposed prediction model. The experimental setup is shown in Figure 10. The milling machine used in the experiment for machining the sample workpiece was a DTX850 CNC (Figure 10b). The cut tool is a face milling cutter with a diameter of 50 mm, and the milling insert model is APMT1604R0.8, as detailed in Table 2. The machining experiments are conducted under three groups of cutting parameters, as listed in Table 3. For each group of cutting parameters, three cutting tools are used for machining. Each tool is considered unusable after machining 168 cutting segments, and the tool wear value exceeds 0.3 mm. Thus, 168 cutting times are used in the run-to-fail experiment of a new tool. A one-cut segment represents the machining length along the short side of the assembly interface. During machining, the three-axis vibration signal of the accelerometer on the spindle and spindle box, the current signal of spindle controller, and the sound pressure signal are collected by a data acquisition instrument at 10 kHz.

Based on prior knowledge, the following signals, vibration, noise, and spindle load, are sensitive to tool wear (i.e., the amplitude of the signals increases with tool wear) and are, thus, good indicators for tool wear and, further, for surface roughness. On the other hand, vibration, noise, and current signals are cheap to collect since the sensors are economical and convenient to install. Therefore, this paper chooses these physical quantities to develop the surface roughness prediction model. It should be noted that cutting force is most directly related to tool wear and surface roughness and is used by many researchers. However, a cutting force sensor has many restrictions and is impractical in real applications, such as the very inconvenient installation for the large workpiece and high cost, and, thus, this paper abandoned the cutting force signal acquisition. After each cutting segment is machined, a portable microscope, Dino-Lite AM7115MZT, is used to measure the tool wear value (*VB*), and another instrument, ISR-C300, is used to measure the surface roughness (*Ra*). The *VB* value of the cutting tool’s flank wear is measured by the ruler of the measurement software provided by the portable microscope, Dino-Lite AM7115MZT as shown in Figure 10b. The *Ra* value is obtained by calculating the average of the surface roughness values of three points in the Z-axis cutting direction of the spindle in each cutting segment, as shown in Figure 10c. The time-domain waveform of the sensor data is shown in Figure 11. It can be found that with the increase of the cutting number, the amplitude of the sensor value, such as sound pressure, vibration, and current, is gradually increased.

### 4.2. Experimental Data Preparation

To verify the proposed framework for surface roughness prediction integrating tool wear under variable cutting parameters, six transfer tasks, T1–T6, are designed, as summarized in Table 3. Under the column “Direction of transfer task,” “P1→P2” means that the model trained under the cutting parameter of group P1 (source domain) is transferred to predict the surface roughness under the cutting parameters of group P2 (target domain). As listed in Table 4, the data of corresponding tool numbers are given in the training and test data. “Tool-A” and “Tool-B” represent the first and second tools during the test stage, respectively.

According to the transfer learning strategy detailed in Section 3.3.1, the number of samples used in the four stages during the training is summarized in Table 5. The first stage uses the source domain samples to train the feature extraction module, SAE. The second stage also employs the source domain samples to pre-train the basic prediction model, SAE–LSTM. The third stage utilizes some source domain samples and a proportion of unlabeled target domain samples to fine-tune the feature extraction module. The fourth stage uses a small amount of label data from the target domain (10 samples, accounting for 2% of the total target domain samples) to fine-tune the prediction module of the SAE–LSTM. Note that the source domain samples are all labeled, and the samples of the surface roughness prediction model at *t* are *X*_*Ra_t_*_−4_, *X*_*Ra_t_*_−3_, *X*_*Ra_t_*_−2_, *X*_*Ra_t_*_−1_, and *X*_*Ra_t_*. Finally, the rest of the unlabeled samples of the target domain are used to test the prediction model.

### 4.3. Results and Discussion

#### 4.3.1. Model Parameter Setting

The hyper-parameter settings, as well as the output shape of each layer of the tool wear prediction model and surface roughness prediction model, are as shown in Table 6.

#### 4.3.2. Optimal Parameter Selection for the Surface Roughness Prediction Model

During the training process, the following two parameters are found to have a considerable effect on the performance of the surface roughness prediction model: (1) the number of labeled samples of the target domain used for fine-tuning at the 4th stage of transfer learning; and (2) the *k*, in Equations (4) and (5) (i.e., the length of historical data used as the input of the prediction model). Therefore, analyzing these two parameters and selecting the optimal values are necessary.

(1)Prediction effect using different numbers of fine-tuning samples

Consider transfer task T1 as an example. The prediction performance is investigated by adjusting the number of fine-tuning samples to observe the *RMSE* of the prediction results, as shown in Figure 12. The *RMSE* is observed to significantly decrease with the increase in the number of fine-tuning samples. When the number of fine-tuning samples reaches 10, any further increase in this number only has a slight effect on reducing the *RMSE*. Similar conclusions are reached in transfer tasks T2–T6. Given that collecting labeled data from the target domain is difficult in practice, 10 samples are used to fine-tune the surface roughness prediction model.

(2)Effect of using different lengths of historical data (*k*) on prediction

The resulting variations of *RMSE* with *k* (*k* vs. *RMSE* curve) and training time with *k* (*k* vs. *training time* curve) when transfer task T1 is considered and the number of fine-tuning samples is fixed to 10 are shown in Figure 13. The blue and orange lines are the *k* vs. *RMSE* curves of CT2-2 and CT2-3, respectively. The green line is the *k* vs. *training time* curve of each epoch in the training process. The *RMSE* is observed to distinctly decrease with the increase in *k* until *k* = 5. Beyond this, any further increase in *k* has an extremely slight effect on reducing the *RMSE*, causing *RMSE* to fluctuate; in addition, the training time rapidly rises with *k*. By considering the *RMSE* and training time, the selected *k* value is 5.

#### 4.3.3. Results of Surface Roughness Prediction with Variable Cutting Parameters

To explore the advantages of some key training stages and modules of the proposed method, ablation studies were implemented. Different training stages and modules of the proposed models are combined to form new prediction models; next, their performance is evaluated. The following four models have been designed for comparison: (1) proposed model; (2) model without LSTM module; (3) model without transfer learning (without TL); and (4) model disregarding tool wear (without VB). In addition, this paper uses *RMSE* and *MAPE* as the metrics to compare the performance of the proposed surface roughness prediction model with the ones in [32,56,57,58]. The offline measured value is considered as the real ground truth, which serves as the benchmark of the online prediction values. Moreover, the measurement data are susceptible to the random factors of the machined surface texture, causing the surface roughness to fluctuate rather than monotonically increase. However, the overall trend of measured roughness values is observed to increase with the number of cuts.

The comparison of the four models in all six transfer tasks is quantified by computing the *RMSE*, as summarized in Table 7 and visualized in Figure 14. The proposed method outperforms the three other models, especially the “Without TL” model in all of the six transfer tasks. This indicates that adopting transfer learning effectively resolves the problem of surface roughness prediction under variable cutting parameters. 

In addition, in the six transfer tasks, the maximum *RMSE* of the proposed method is 0.089 μm, while the *RMSE* of the best model reported in the literature [56] is 0.2652 μm. The maximum *RMSE* of this paper is 33.6% of the *RMSE* in literature [56]. Furthermore, as summarized in Table 8, the *MAPE* of the proposed method in the six transfer tasks is less than the minimum *MAPE* reported in the literature [32,57,58]. In the literature [32], the features of the first class in the input layer of the surface roughness prediction model include the cutting parameters, which are feed per tooth, cutting depth, clamping torque of vise, and the removed volume accumulation per cutter. The features extracted from the vibration signals were selected in the second class if their correlation coefficients to *Ra* values have absolute values higher than 0.4. The third class consists of the features of the cutting parameters as well as the selected features of the second class. Literature [58] analyzed *MAPE* of the models of novel adaptive neuro-fuzzy inference system (NANFIS), standard ANFIS (SANFIS), complex ANFIS (CANFIS), and standard ANFIS trained by improved particle swarm optimization (IANFIS).

The prediction results of the four models on CT2-2 of task T1 and on CT3-2 of task T2 are illustrated in Figure 15 and Figure 16, respectively. These results show that the proposed model outperforms the other models in both T1 and T2 tasks and tracks the true surface roughness with time well. The prediction result of the “Without VB” model indicates the trend and considerable fluctuation of surface roughness. In contrast, because the “Without TL” model accounts for the tool wear, the fluctuations are considerably smaller. However, without transfer learning, the predicted roughness values by the “Without TL” model significantly deviate from the ground truth roughness values once the cutting parameters change. Therefore, the proposed model can integrate tool wear and sensor data to achieve an accurate online surface roughness prediction.

According to the results in Figure 15 and Figure 16 and Table 7 and Table 8, the proposed method can better predict the surface roughness than the other methods and can be applied to a relatively large parameter range of assembly interface machining. Although the proposed method requires experimental data collection and processing, it can provide a reference solution for the online surface quality monitoring of the assembly interface machining process to avoid offline measurement and relying on worker experience.

## 5. Conclusions and Future Work

The machined surface quality of assembly interfaces is critical to the final quality of an aircraft. Therefore, the online monitoring of the quality of the assembly interfaces of machined surfaces can ensure that the appropriate machining parameters are adjusted, and machining quality is achieved. This paper proposes a transfer-learning-based surface roughness prediction method for assembly interfaces considering tool wear variation. The proposed method can solve the surface roughness online prediction problem under variable cutting parameters. The proposed method is validated by a machining experiment on the assembly interface of the vertical tail of a large passenger aircraft. The contributions of this study are as follows: (1)The surface roughness of an assembly interface is predicted by integrating the time-varying characteristics of sensor data and tool wear. This enables the prediction of surface roughness to more accurately reflect the influence of the time-varying machining process on surface roughness;(2)The multi-time-step input SAE is designed to extract the time-series features of raw input data. It is combined with the LSTM to predict the surface roughness degradation in assembly interface machining;(3)The proposed SAE–LSTM prediction model adopts transfer learning, which can solve the problem encountered in the online prediction of surface roughness for assembly interfaces under variable cutting parameters;(4)Ablation studies are implemented by conducting a machining experiment on the assembly interfaces of a titanium alloy vertical tail. The proposed method can predict the surface roughness of assembly interfaces under variable cutting parameters and achieve high prediction accuracy. It provides supporting data and theory for the online monitoring of surface roughness and adjustment of cutting parameters in assembly interface machining. This method can also serve as a reference for the online monitoring of assembly interface machining of spacecraft, wind power equipment, etc.

Currently, the proposed method still requires a small amount of label data to fine-tune the target domain model. In the future, the complexity of modeling can be reduced by automatic labeling. The proposed method is validated on only one part of the assembly interface of a vertical tail. In practice, the machining process involves multiple assembly interfaces, and the experimental conditions may be different; hence, this method will be applied and verified in practical scenarios in the future. There are many other surface texture parameters for evaluating surface quality, such as 3-D surface roughness, surface waviness, surface form, etc. In the future, online monitoring of more parameters should be studied to comprehensively evaluate surface quality. Further, the means for solving the problem of surface roughness prediction considering different tools and machining materials will be investigated. Finally, the method of predicting surface roughness by a hybrid model that combines the strong interpretable model (numerical method, cutting dynamics method, etc.) and artificial intelligence model is worthy of in-depth study.

## Figures and Tables

**Figure 1 sensors-22-01991-f001:**
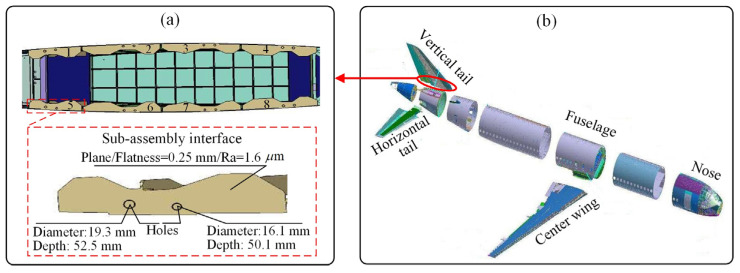
The assembly interfaces of an aircraft vertical tail: (**a**) Assembly interfaces; (**b**) large aircraft.

**Figure 2 sensors-22-01991-f002:**
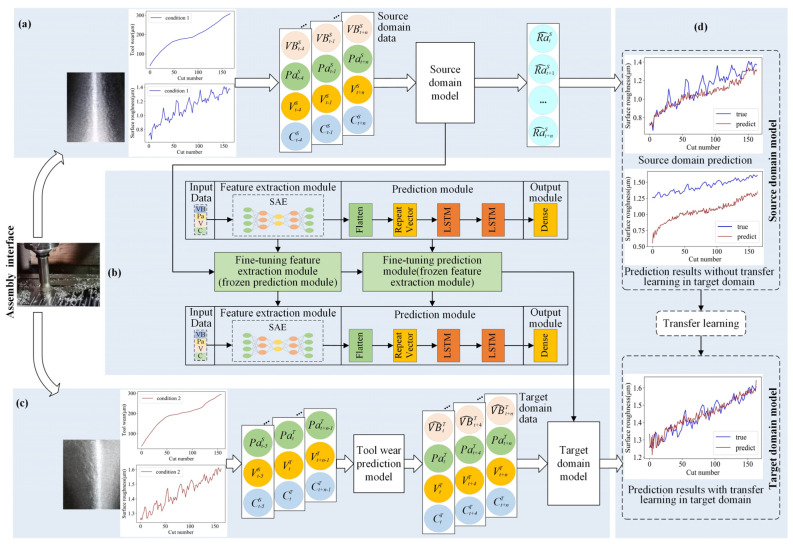
Surface roughness prediction framework for the assembly interface: (**a**) Pre-training of the surface roughness prediction model in the source domain; (**b**) transfer learning for the modules of the source domain model; (**c**) prediction of surface roughness in the target domain; (**d**) surface roughness prediction.

**Figure 3 sensors-22-01991-f003:**
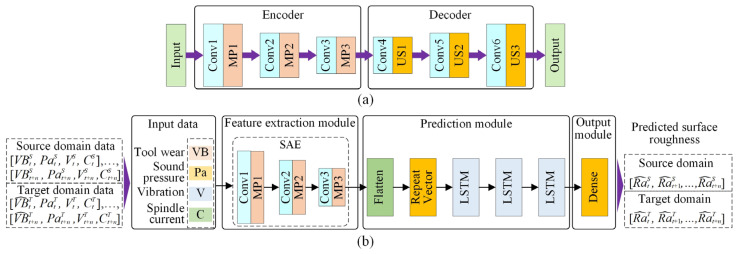
SAE and surface roughness prediction model structure based on SAE–LSTM: (**a**) SAE; (**b**) the surface roughness prediction model structure based on SAE-LSTM.

**Figure 4 sensors-22-01991-f004:**
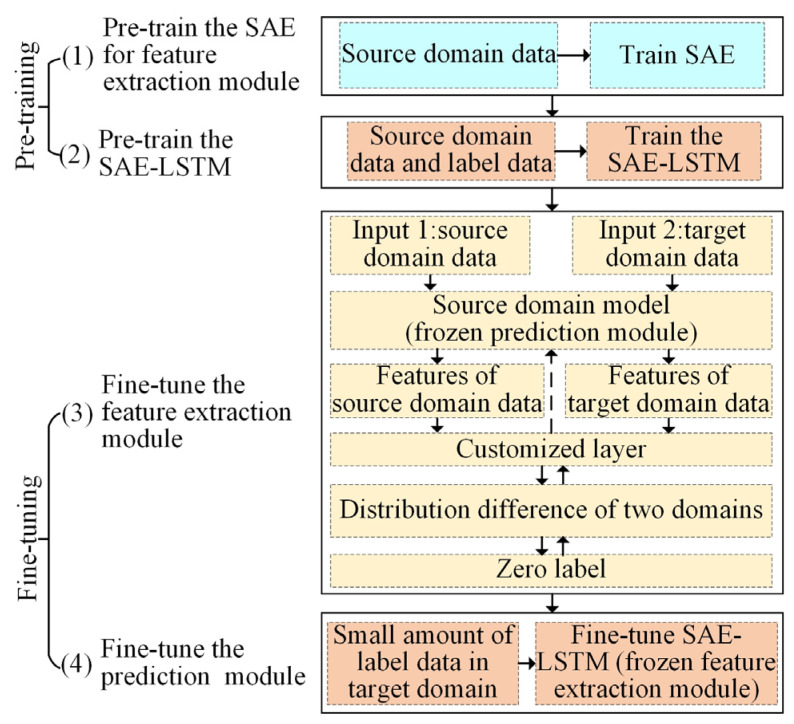
Transfer learning strategy based on a multi-stage model training process.

**Figure 5 sensors-22-01991-f005:**
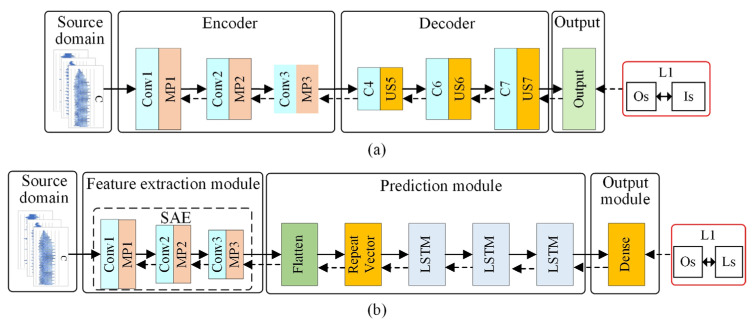
Schematic of pre-training: (**a**) SAE pre-training; (**b**) SAE–LSTM pre-training.

**Figure 6 sensors-22-01991-f006:**
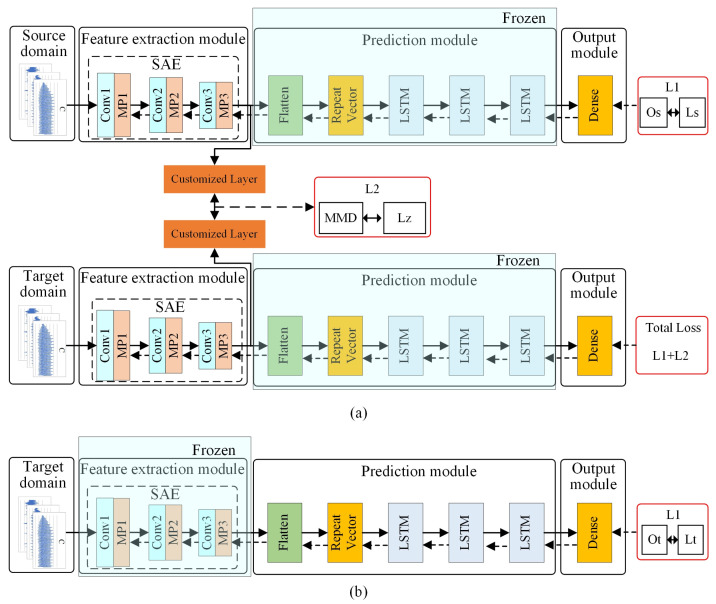
Schematic of fine-tuning: (**a**) Fine-tuning of the feature extraction module; (**b**) fine-tuning of the prediction module.

**Figure 7 sensors-22-01991-f007:**
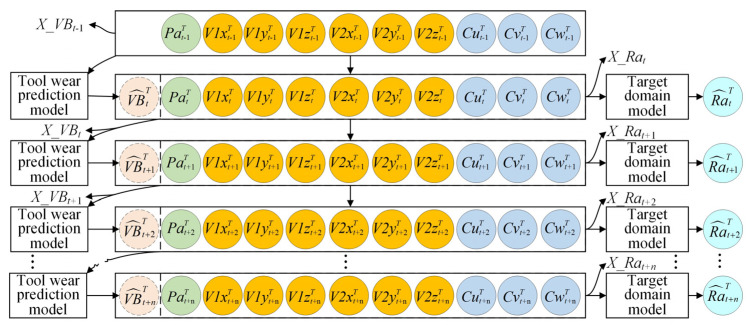
Surface roughness prediction integrating tool wear.

**Figure 8 sensors-22-01991-f008:**
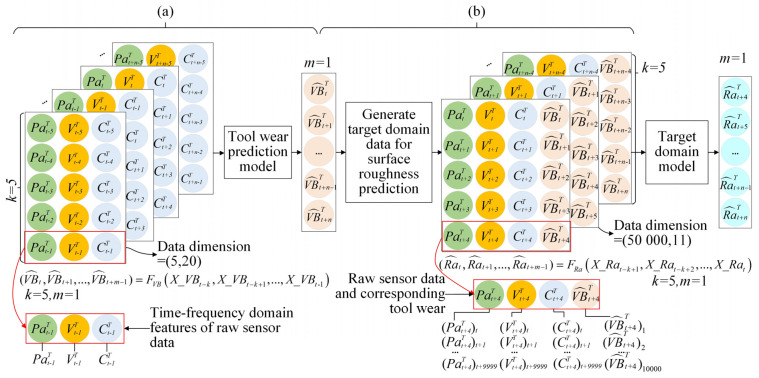
Data preparation for surface roughness prediction: (**a**) Tool wear prediction process in the target domain; (**b**) data generation process for roughness prediction integrating tool wear in the target domain.

**Figure 9 sensors-22-01991-f009:**
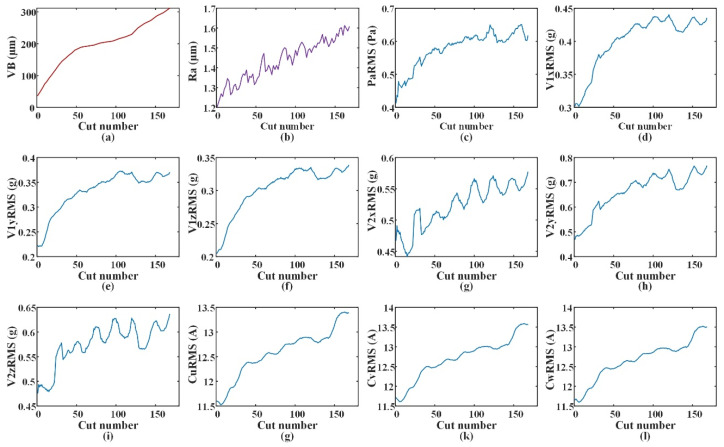
*VB*, *Ra*, and features of sensor data: (**a**) Tool wear; (**b**) surface roughness; (**c**) RMS of sound pressure signal; (**d**) RMS of vibration signals in *x*-axis on the spindle; (**e**) RMS of vibration signals in *y*-axis on the spindle; (**f**) RMS of vibration signals in *z*-axis on the spindle; (**g**) RMS of vibration signals in *x*-axis on the spindle box; (**h**) RMS of vibration signals in *y*-axis on the spindle box; (**i**) RMS of vibration signals in *z*-axis on the spindle box; (**g**) RMS of u-phase current signal of the spindle controller; (**k**) RMS of v-phase current signal of the spindle controller; (**l**) RMS of w-phase current signal of the spindle controller.

**Figure 10 sensors-22-01991-f010:**
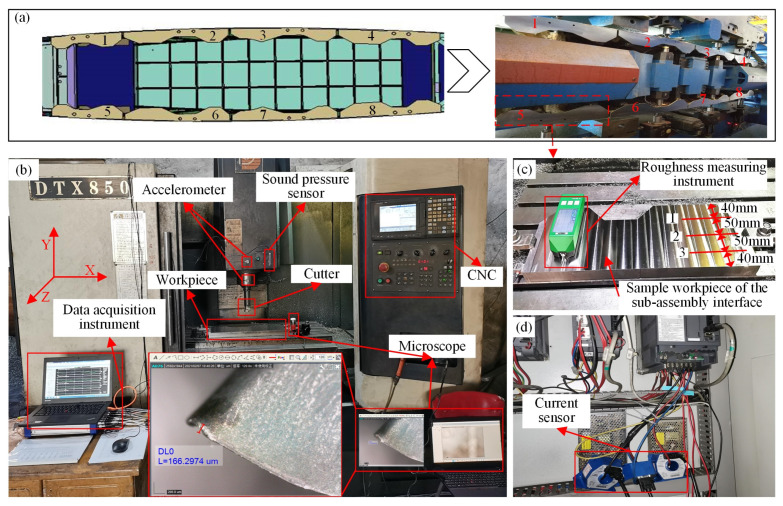
Experimental setup: (**a**) Assembly interfaces; (**b**) machining site; (**c**) sample workpiece; (**d**) spindle controller.

**Figure 11 sensors-22-01991-f011:**
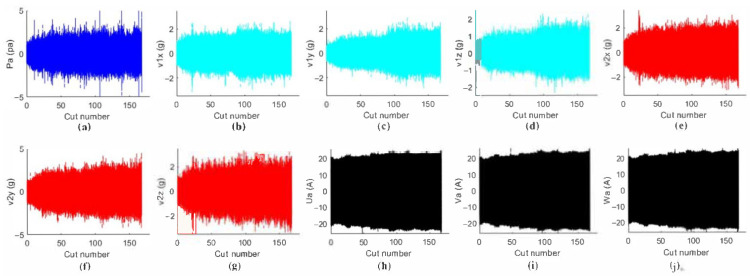
Time-domain waveform of the sensor data: (**a**) Sound pressure signal; (**b**) vibration signals in *x*-axis on the spindle; (**c**) vibration signals in *y*-axis on the spindle; (**d**) vibration signals in *z*-axis on the spindle; (**e**) vibration signals in *x*-axis on the spindle box; (**f**) vibration signals in *y*-axis on the spindle box; (**g**) vibration signals in *z*-axis on the spindle box; (**h**) the u-phase current signal of the spindle controller; (**i**) the v-phase current signal of the spindle controller; (**j**) the w-phase current signal of the spindle controller.

**Figure 12 sensors-22-01991-f012:**
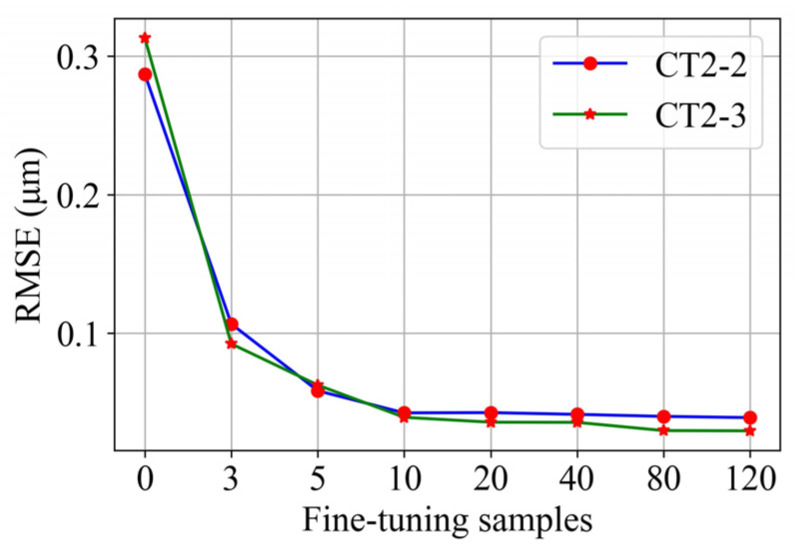
Effect of using different numbers of fine-tuning samples in the T1 task on prediction.

**Figure 13 sensors-22-01991-f013:**
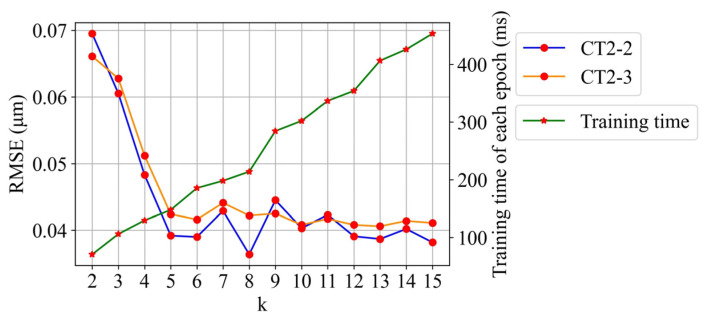
Effect of using different lengths of historical data (*k*) in the T1 task on prediction.

**Figure 14 sensors-22-01991-f014:**
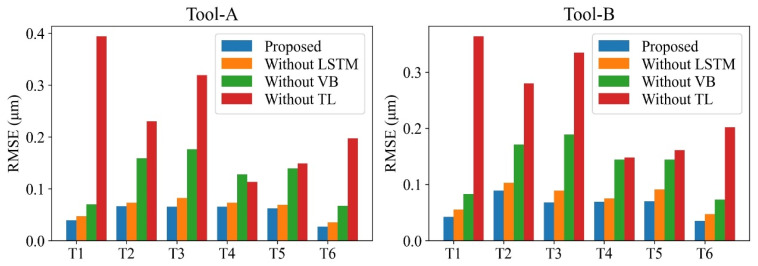
Prediction results considering each transfer task under different models.

**Figure 15 sensors-22-01991-f015:**
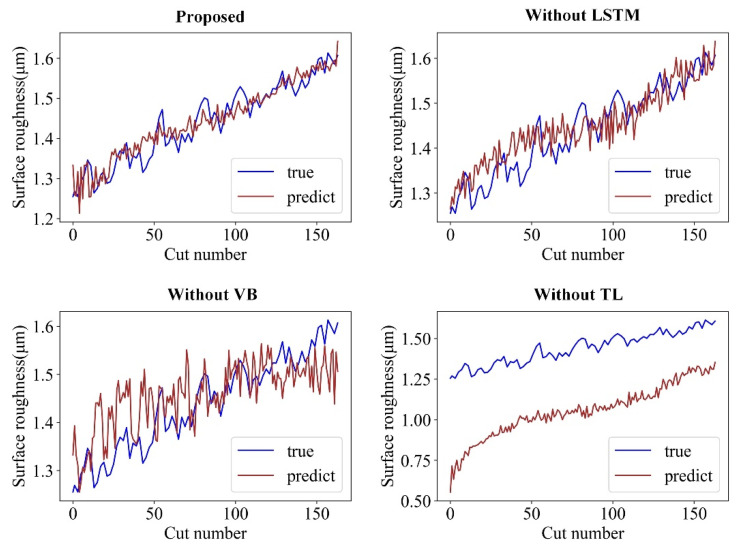
Prediction effect of different models on CT2-2 of the T1 task.

**Figure 16 sensors-22-01991-f016:**
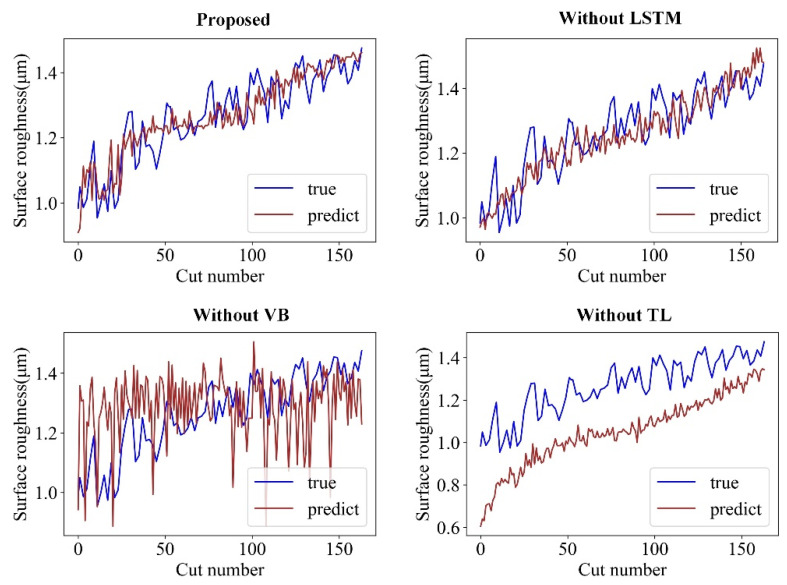
Prediction effect of different models on CT3-2 under the T2 task.

**Table 1 sensors-22-01991-t001:** Definition of terms.

NO.	Term	Explanation
1	Source domain	Machining conditions with complete data (e.g., sensor data, tool wear data, surface roughness data, and cutting parameters)
2	Target domain	Machining conditions that differ from the cutting parameters of the source domain and have incomplete data (e.g., lack of tool wear data and surface roughness data)
3	Source domain data	Data collected under the source domain (e.g., data of tool wear, sound pressure sensor, accelerometer, and current sensor)
4	Target domain data	Data collected under the target domain (e.g., data of tool wear, sound pressure sensor, accelerometer, and current sensor)
5	Source domain model	Surface roughness prediction model trained with source domain data
6	Target domain model	Surface roughness prediction model trained with transfer learning
7	Variable cutting parameters	Different cutting parameters of machining assembly interface in the source and target domains
8	Ground truth labels	Surface roughness values of assembly interface that are measured offline by a contact device under the source or target domain

**Table 2 sensors-22-01991-t002:** Cutting tool parameters.

Cutter Material	Cutter Diameter, *D*, mm	The Number of Inserts, *Nz*	Cutting Edge Radius, *r*, mm	Back Angle, α, °
Carbide	50	4	0.8	11

**Table 3 sensors-22-01991-t003:** Three groups of cutting parameters used in assembly interface machining experiment.

Group no. of Cutting Parameters	Feed Rate, *f* (mm/min)	Axial Depth of Cut, *a_p_* (mm)	Spindle Speed, *n* (r/min)	No. of Tested Cutting Tools
P1	210	0.5	450	CT1-1–CT1-3
P2	290	0.5	500	CT2-1–CT2-3
P3	250	0.5	380	CT3-1–CT3-3

**Table 4 sensors-22-01991-t004:** Training and test data settings under different transfer tasks.

Transfer Task No.	Direction of Transfer Task (Source Domain→Target Domain)	Training Data	Test Data	
			Tool-A	Tool-B
T1	P1→P2	CT1-1–CT1-3, CT2-1	CT2-2	CT2-3
T2	P1→P3	CT1-1–CT1-3, CT3-1	CT3-2	CT3-3
T3	P2→P1	CT2-1–CT2-3, CT1-1	CT1-2	CT1-3
T4	P2→P3	CT2-1–CT2-3, CT3-1	CT3-2	CT3-3
T5	P3→P1	CT3-1–CT3-3, CT1-1	CT1-2	CT1-3
T6	P3→P2	CT3-1–CT3-3, CT2-1	CT2-2	CT2-3

**Table 5 sensors-22-01991-t005:** Data sample size at each stage.

Task	Training				Testing
	Stage 1 (Number of Source Domain Samples)	Stage 2 (Number of Source Domain Samples)	Stage 3 (Number of Source Domain/Number of Unlabeled Target Domain Samples)	Stage 4 (Labeled Target Domain Samples)	Test Stage Samples
T1	164 samples of CT1-1	164 samples of CT1-2	164 samples of CT1-3/164 samples of CT2-1	10 samples of CT2-1	328
T2	164 samples of CT1-1	164 samples of CT1-2	164 samples of CT1-3/164 samples of CT3-1	10 samples of CT3-1	328
T3	164 samples of CT2-1	164 samples of CT2-2	164 samples of CT2-3/164 samples of CT1-1	10 samples of CT1-1	328
T4	164 samples of CT2-1	164 samples of CT2-2	164 samples of CT2-3/164 samples of CT3-1	10 samples of CT3-1	328
T5	164 samples of CT3-1	164 samples of CT3-2	164 samples of CT3-3/164 samples of CT1-1	10 samples of CT1-1	328
T6	164 samples of CT3-1	164 samples of CT3-2	164 samples of CT3-3/164 samples of CT2-1	10 samples of CT2-1	328

**Table 6 sensors-22-01991-t006:** The hyper-parameter settings of the tool wear prediction model and surface roughness prediction model.

Tool Wear Prediction Model				Surface Roughness Prediction Model							
Layer	Symbol	Activation Function	Output Shape	Layer	Symbol	Activation Function	Output Shape	Layer	Symbol	Activation Function	Output Shape
1	Input	/	(5,20)	1	Input	/	(50,000,11)	10	LSTM	ReLU	(1,64)
2	LSTM	ReLU	(5,32)	2	Conv1D	ReLU	(50,000,16)	11	LSTM	ReLU	(1,32)
3	LSTM	ReLU	(5,128)	3	MaxPooling1D	/	(2500,16)	12	LSTM	ReLU	(1,16)
4	Flatten	/	(640)	4	Conv1D	ReLU	(2500,8)	13	Dense	/	(1,1)
5	Repeat Vector	/	(1,640)	5	MaxPooling1D	/	(250,8)				
6	LSTM	ReLU	(1,128)	6	Conv1D	ReLU	(250,4)				
7	LSTM	ReLU	(1,64)	7	MaxPooling1D	/	(25,4)				
8	LSTM	ReLU	(1,32)	8	Flatten	/	(100)				
9	Dense	/	(1,1)	9	Repeat Vector	/	(1,100)				

**Table 7 sensors-22-01991-t007:** Comparison results of different prediction models.

Task	Tool-A (*RMSE*)				Tool-B (*RMSE*)			
	Proposed	Without LSTM	Without VB	Without TL	Proposed	Without LSTM	Without VB	Without TL
T1	0.039	0.047	0.070	0.394	0.042	0.055	0.083	0.364
T2	0.066	0.073	0.159	0.230	0.089	0.103	0.171	0.280
T3	0.065	0.082	0.176	0.319	0.068	0.089	0.189	0.335
T4	0.065	0.073	0.128	0.113	0.069	0.075	0.144	0.148
T5	0.062	0.069	0.139	0.149	0.070	0.091	0.144	0.161
T6	0.027	0.035	0.067	0.197	0.035	0.047	0.073	0.202

**Table 8 sensors-22-01991-t008:** *MAPE* of all six transfer tasks and *MAPE* of [32,57,58].

Proposed			[32]		[57]		[58]	
Task	Tool-A	Tool-B						
T1	1.82%	2.76%	Features of first class	29%	1-D CNN	8.92%	NANFIS	6.2%
T2	4.04%	5.13%					IANFIS	6.2%
T3	4.65%	5.17%	Features of second class	25%	FFT-DNN	8.35%	SANFIS	19.9%
T4	4.94%	4.90%					CANFIS	25.6%
T5	4.71%	4.89%	Features of third class	18%	FFT-LSTM	6.57%		
T6	1.56%	2.0%						

## Data Availability

Not applicable.

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
