# Peer review of "Online Surface Roughness Prediction for Assembly Interfaces of Vertical Tail Integrating Tool Wear under Variable Cutting Parameters"

_sensors, 2022, doi:10.3390/s22051991_

Round 1

Reviewer 1 Report

The study titled ``Online surface roughness prediction for assembly interfaces of vertical tail integrating tool wear under variable cutting parameters`` examines wear that occur under variable cutting parameters with online prediction. The study is quite original and important, but some corrections need to be made.

More information can be given about the results obtained in the abstract.

Introduction and Abstract's introductory sentences are the same, can it be better expressed rather than similarity?

The literature used can be increased when comparing the online and offline approach.

2.Related Works title is not needed, grouping can be done in Introduction under 1.1.Surface roughness prediction with indirect sensors and 1.2.Surface roughness prediction with direct sensors”. In addition, what is told in the title number 2 is irrelevant from the introduction.

4.3.3. More literature should be presented for the results obtained in the title.

Author Response

Response to Reviewer 1 Comments

Point 1: More information can be given about the results obtained in the abstract.

Author response 1:

Thanks for the reviewer's comments. We added more information about the predicted results in the abstract section.

Author action 1:

We added the following text in the abstract.

(Abstract) “Specifically, the minimum values of root mean square error and mean absolute percentage error of the prediction results after transfer learning are 0.027μm and 1.56%, respectively.

Point 2: Introduction and Abstract's introductory sentences are the same, can it be better expressed rather than similarity?

Author response 2:

Thanks for the reviewer's comments. We have modified the expression of the Abstract part, hoping that this can better express the background of this paper.

Author action 2:

We revised the abstract's introductory sentences as follows.

(Abstract) “Monitoring surface quality during machining has considerable practical significance for the performance of high-value products, particularly for their assembly interfaces.

Point 3: The literature used can be increased when comparing the online and offline approach.

Author response 3:

Thanks for the valuable comments. We added the literature on the offline and online measurement of surface roughness and revised the description of the corresponding paragraphs of the article.

Author action 3:

We revised the second paragraph of Section 1 as below and added more literature on the approaches regarding offline and online surface roughness measurement.

(2nd paragraph, Section 1) “Surface roughness is typically monitored either offline or online. The offline approach involves the use of contact measurement equipment [13, 14] and optical measurement equipment [15-17]. The offline approach has drawbacks such as long measurement time, specific requirements to the working environment, complexity to set up, etc. In contrast, online methods are not hindered by the foregoing disadvantages. In view of this, they have attracted increasing research interest [18–20]. Online methods aim to establish a mapping model (prediction model) between surface roughness and online monitoring data, such as cutting parameters and sensor signals [21, 22]. In addition, there are also studies on surface roughness estimation through the numerical method, which is difficult to integrate the real-time sensor data of the machining site to adaptively adjust the roughness prediction online. Given the cons of offline and numerical methods, this paper uses the scheme for surface roughness online prediction based on real-time sensor data.

[14] Dhanasekar, B., Ramamoorthy, B. Restoration of blurred images for surface roughness evaluation using machine vision. Tribol. Int. 2010, 43, 268-276.

[15] Laopornpichayanuwat, W., Visessamit, J., Tianprateep, M. 3-D Surface roughness profile of 316-stainless steel using vertical scanning interferometry with a superluminescent diode. Measurement. 2012, 45, 2400-2406.

[16] Nwaogu, U.C., Tiedje, N. S., Hansen, H. N. A non-contact 3D method to characterize the surface roughness of castings. J. Mater. Process. Technol. 2013, 213, 59-68.

[17] SamtaÅŸ, G. Measurement and evaluation of surface roughness based on optic system using image processing and artificial neural network. Int. J. Adv. Manuf. Technol. 2014, 73, 353–364.

[22] Maher, I., Eltaib, M.E.H., Sarhan, A.A.D., Zahry, R.M.E. Cutting force-based adaptive neuro-fuzzy approach for accurate surface roughness prediction in end milling operation for intelligent machining. Int. J. Adv. Manuf. Technol. 2015, 76, 1459-1467.

Point 4: Related Works title is not needed, grouping can be done in Introduction under 1.1.Surface roughness prediction with indirect sensors and 1.2.Surface roughness prediction with direct sensors”. In addition, what is told in the title number 2 is irrelevant from the introduction.

Author response 4:

We would like to thank the reviewer for this conscientious comment. This paper belongs to the category of online surface roughness prediction. The reason why we summarize the literature in an independent section (that is, Section 2 related works) is that the research on surface roughness online prediction is very rich and versatile. If the literature are only reviewed in the introduction, the content of “Introduction” will be too long. Therefore, we explain the research motivation of this paper in the introduction, summarize the limitations of existing online prediction methods of surface roughness in the related work, and put forward the research scope, problems, and research contributions of this paper. In addition, we noticed that separating literature review from "Introduction" is also a common style for writing a scientific paper and is used by many researchers. So we decided to retain the Related Works section, for this, we added a paragraph at the beginning of Section 2 “Related Works” to improve the logic.

Author action 4:

We added a paragraph at the beginning of Section 2 “Related Works” to improve the structural logic of this section, as shown below.

(1st paragraph, Section 2) “In this section, firstly, surface roughness online prediction methods are reviewed as indirect and direct sensor-based techniques depending on whether or not surface roughness is directly measured. Then, the advantages and disadvantages of the current online surface roughness prediction methods are summarized. Finally, the problems solved in this paper and specific contributions are given.

In addition, we added the following text in subsection 2.3 “Summary and analysis”, to further strengthen the motivation of this study, after reviewing the summarizing the literature.

(1st paragraph, Section 2.3) “The proposed method belongs to the category of artificial intelligence model, which can make full use of real-time sensor data to drive the model to predict surface roughness online.

Point 5: 4.3.3. More literature should be presented for the results obtained in the title.

Author response 5:

Thanks for the reviewer’s valuable comment. We added two pieces of literature that are most relevant to the theme and results of this paper and compared the MAPE indicator to prove the effectiveness and the advanced nature of the method proposed in this paper.

Author action 5:

We added literature in section 4.3.3, updated table 8, and added the following text to analyze the results.

(3rd paragraph, Section 4.3.3) “Furthermore, as summarized in Table 8, the MAPE of the proposed method in the six transfer tasks is less than the minimum MAPE reported in the literature [33], [58], and [59]. In literature [33], the features of the first class in the input layer of surface roughness prediction model include the cutting parameters, which are feed per tooth, cutting depth, clamping torque of vise, and the removed volume accumulation per cutter. The features extracted from the vibration signals were selected in the second class if their correlation coefficients to Ra values have absolute values higher than 0.4. The third class consists of the features of the cutting parameters as well as the selected features of the second class. Literature [59] analyzed MAPE of the models of novel adaptive neuro fuzzy inference system (NANFIS),standard ANFIS (SANFIS), complex ANFIS (CANFIS), and standard ANFIS trained by improved particle swarm optimization (IANFIS).

Table 8. MAPE of all six transfer tasks and MAPE of [33], [58], and [59].

Proposed

[33]

[58]

[59]

Task

Tool-A

Tool-B

T1

1.82%

2.76%

Features of

first class

29%

1-D CNN

8.92%

NANFIS

6.2%

T2

4.04%

5.13%

IANFIS

6.2%

T3

4.65%

5.17%

Features of

second class

25%

FFT-DNN

8.35%

SANFIS

19.9%

T4

4.94%

4.90%

CANFIS

25.6%

T5

4.71%

4.89%

Features of

third class

18%

FFT-LSTM

6.57%

T6

1.56%

2.0%

[58]Lin, W.J., Lo, S.H., Young, H.T., Hung, C.L. Evaluation of Deep Learning Neural Networks for Surface Roughness Prediction Using Vibration Signal Analysis. Appl. Sci. 2019, 9,1462.

[59]Xu, L.H., Huang, C. Z., Niu, J.H., Wang, J., et al. Prediction of cutting power and surface quality, and optimization of cutting parameters using new inference system in high-speed milling process. Adv. Manuf. 2021. https://doi.org/10.1007/s40436-020-00339-6.

Reviewer 2 Report

The paper presents some interesting and innovative research work on surface roughness prediction for assembly interfaces of the vertical tail integrating tool wear under variable cutting process parameters. The work is further supported by analysis and experimental results. However, the paper manuscript should undertake the following revisions in order to reach its publication at the journal:

(1) In section 3, the paper should should use a paragraph(s) to provide a further clarification and discussion on the underlying machining dynamics and mechanics affecting the surface roughness generation, e.g. dynamic cutting forces and cutting dynamics vs tool wear vs surface roughness generation. 

(2) In section 4, the manuscript should provide details of the cutting tool, such as tool material, tool geometry, cutting edge radius, etc., which are essentially important in affecting the cutting mechanics and dynamics and consequently the tool wear and surface generation.

(3) In the work presented, the surface roughness is defined as Ra or equivalent. However, for the assembly interfaces of the vertical tail component, 3D surface roughness parameters and the associated functionality can be very important and better defined for the engineering purpose. This should be briefly reviewed in Introduction section.

(4) The following very relevant paper and book are recommended to be included in References section, particularly against the above comments (3), (1) and (2):

  • Characterization of the surface functionality on precision machined engineering surfaces, International Journal of Advanced Manufacturing Technology, Vol.38, No.3-4, 2008, 402-409.
  • Machining Dynamics: Theory, Applications and Practices, Springer, London, November 2008.

Author Response

Response to Reviewer 2 Comments

Point 1: In section 3, the paper should use a paragraph(s) to provide a further clarification and discussion on the underlying machining dynamics and mechanics affecting the surface roughness generation, e.g. dynamic cutting forces and cutting dynamics vs tool wear vs surface roughness generation.

Author response 1:

We would like to thank the reviewer for this conscientious and beneficial comment. We agree with the reviewer that it is very meaningful to elaborate on the machining dynamics and mechanism affecting the surface roughness generation. After carefully reading the two literature recommended by the reviewer, we found that the surface generation model in section 7.4 of literature [47] has a certain reference value for the surface roughness generation mechanism, such as Eq. (1), where Xi is the radial coordinate of the ith tooth of the tool in machining, Yi is the feed direction coordinate of the ith tooth in machining, Ri is the actual cutting radius of the ith tooth, φis the helix lag angle,  ft  is the feedrate, m is the teeth number of the tool, is the spindle rotation angular velocity, xc, yc and xw, yw are the regenerative displacements of the tool and workpiece, respectively. The formula explains that the tool wear will affect the tool radius, and then affect the machined surface morphology. In addition, we also read literature [48], in which the cutting force model of the worn tool is proposed, as shown in Eq. (2) - (3), where Ftw and Frw are the friction and squeezing force caused by the worn tool, respectively.  τ0 and σ0 are the constant values of shear stress and normal stress, respectively. x is the distance from a certain point on the flank surface to the cutting edge and VBp is the width of the plastic flow zone. These two formulas explain that the unstable cutting force will be produced after the wear of each cutting edge. Therefore, the above two models will affect the surface roughness.

However, there will be stress, chatter, temperature, etc. [49-51] complex phenomena in the machining with worn tools. Only the above two models can not provide a comprehensive and direct surface roughness dynamic model to explain the influence of worn tools on surface roughness in dynamic machining. The purpose of this paper is to express the influence of uncertain factors on surface roughness by integrating tool wear with sensor data and establishing a surface roughness prediction model based on this. Therefore, for the integrity of the paper and the fluency of reading, we consider not adding a dynamics formula to explain the mechanism of tool wear and its effect on surface roughness. However, we still believe that it is necessary to quote the literature recommended by the reviewer and add a paragraph to explain the impact of machining dynamics on the generation of surface roughness, which may have some enlightenment for future research in this field.

Please see the formula in the attachment(1)

Please see the formula in the attachment(2)

Please see the formula in the attachment(3)

Author action 1:

We added a paragraph in section 3.2 to explain the influence of machining dynamic mechanism on surface roughness generation, and added a part of the content in the conclusion and future work.

 (1st paragraph, Section 3.2) “Before establishing the model, it is necessary to discuss the consideration of the model input data in this paper. The research shows that in machining dynamics, tool wear has an impact on the surface morphology generation [47], cutting force [48], contact stress, chatter, temperature [49-51], etc. Therefore, it is essential to comprehensively consider the sensor data that reflect the actual machining situation and the tool wear data.

[47]Cheng, K. Machining Dynamics: Theory, Applications and Practices. Springer: London, 2008; pp. 205-207.

[48]Smithey, D.W., Kapoor, S.G., DeVor, R.E. A new mechanistic model for predicting worn tool cutting forces. Mach. Sci. Technol. 2001, 5, 23–42.

[49]Attanasio, A., Ceretti, E., Giardini, C., Filice, L., Umbrello, D. Criterion to evaluate diffusive wear in 3D simulations when turning AISI 1045 steel. Int. J. Mater. Form. 2008, 1, 495-498.

[50]Moradi, H., Vossoughi, G., Movahhedy, M.R. Bifurcation analysis of nonlinear milling process with tool wear and process damping: Sub-harmonic resonance under regenerative chatter. Int. J. Mech. Sci. 2014, 85, 1-19.

[51]Liang, X., Liu, Z. Tool wear behaviors and corresponding machined surface topography during high-speed machining of Ti-6Al-4V with fine grain tools. Tribol. Int. 2018, 121, 321-332.

(Section 5, Conclusions and future work) “Furthermore, the method of predicting surface roughness by a hybrid model that combines the strong interpretable model (numerical method, cutting dynamics method, etc.) and artificial intelligence model is worthy of in-depth study.

Point 2: In section 4, the manuscript should provide details of the cutting tool, such as tool material, tool geometry, cutting edge radius, etc., which are essentially important in affecting the cutting mechanics and dynamics and consequently the tool wear and surface generation.

Author response 2:

Thanks for the reviewer’s valuable comment. We added the detailed parameter information of the cutting tool.

Author action 2:

We added a table of cutting tool parameters in Section 4.1, as shown below, and updated the description of the cutting tool parameters.

Table 2. Cutting tool parameters

Cutter

Material

Cutter diameter

, D, mm

The number of

inserts, Nz

Cutting edge

radius, r, mm

Back angle

, α, o

Carbide

50

4

0.8

11

Point 3: In the work presented, the surface roughness is defined as Ra or equivalent. However, for the assembly interfaces of the vertical tail component, 3D surface roughness parameters and the associated functionality can be very important and better defined for the engineering purpose. This should be briefly reviewed in Introduction section.

Author response 3:

Thanks for the reviewer’s valuable comment. 3D surface roughness parameters and the associated functionality are indeed very important for evaluating machining quality. The reason for modeling and predicting Ra in this paper is that Ra is an important quality evaluation indicator of the vertical tail assembly interface of a certain type of aircraft involved in the experiment. In addition, if the surface quality needs to be comprehensively evaluated, more parameters are needed, such as 3-D surface roughness parameters, surface waviness, surface form, etc. However, it is beyond the research scope of this paper. We will study the prediction of more parameters in the future

Author action 3:

In Section 1, we added more reviews of surface roughness and explained why we chose surface roughness Ra.

(1st paragraph, Section 1)”There are many parameters affecting surface quality, such as surface roughness (2-D and 3-D surface roughness), surface waviness, surface form, etc. [5, 6], among which, surface roughness parameters and the associated functionality are very important for the evaluation of surface integrity and machining quality [7, 8] since surface roughness significantly influences the assembly accuracy, fatigue strength, corrosion resistance, and contact stiffness of parts [9, 10]. Consequently, surface roughness becomes an important parameter concerned by engineers during aviation manufacturing[11, 12], for instance, in the manufacturing of assembly interface of aircraft vertical, as shown in Figure 1.

[5] Aris, N.F.M., Cheng, K. Characterization of the surface functionality on precision machined engineering surfaces. Int. J. Adv. Manuf. Technol. 2008, 38, 402-409.

[6] Grzesik, W. Advanced Machining Processes of Metallic Materials. Elsevier: Amsterdam, 2008; pp. 405-420.

[11] Wang, M.Y., Yang, C.W., Ziliang, L.I., Zhao, S.F., et al. Effects of surface roughness on the aerodynamic performance of a high subsonic compressor airfoil at low Reynolds number. Chin. J. Aeronaut. 2021, 34, 71–81.

[12] Duan, Z.J., Li, C.H., Zhang, Y.B., Dong, L., et al. Milling surface roughness for 7050 aluminum alloy cavity influenced by nozzle position of nanofluid minimum quantity lubrication. Chin. J. Aeronaut. 2021. 34, 33–53.

Point 4: The following very relevant paper and book are recommended to be included in References section, particularly against the above comments (3), (1) and (2):

Characterization of the surface functionality on precision machined engineering surfaces, International Journal of Advanced Manufacturing Technology, Vol.38, No.3-4, 2008, 402-409.

Machining Dynamics: Theory, Applications and Practices, Springer, London, November 2008.

Author response 4:

We have read the two pieces of literature recommended by the reviewer and added them to the References section, such as the replies from Point1 to Point3.

Reviewer 3 Report

The article is interesting and describes the current methods of predicting phenomena in the cutting process. The author clearly presents the analyzed methods and achievements of other scientists in this field. The article proposes a new method of surface roughness prediction analysis based on tool wear differs under variable cutting parameters. 
However, there are some questions about the article: 
1. There are many works on the study of wear depending on the cutting data. Could these results be used to create a roughness prediction model?

2. Does the proposed method depend on the type of cutting material and have there been any preliminary attempts to determine the impact of the material?  

3. Has the presented method been verified by numerical methods? 

Author Response

Response to Reviewer 3 Comments

Point 1: There are many works on the study of wear depending on the cutting data. Could these results be used to create a roughness prediction model?

Author response 1:

Thanks for the reviewer’s comments. The research objects of tool wear prediction and surface roughness prediction are different, and many factors interfere with surface roughness in machining, which is quite different from tool wear prediction. Therefore, it is not feasible to predict the surface roughness only by using the tool wear prediction model. But indeed, in machining, the contact angle with the workpiece is changed after tool wear, which directly affects the surface roughness. If the influence of tool wear is integrated into the surface roughness prediction, the prediction will be more accordant with the actual machining situation. Therefore, this paper is to attempts to integrate sensor data and tool wear to predict surface roughness.

Author action 1:

We updated the third paragraph of Section 1 to explain the motivation of integrating tool wear to predict surface roughness under variable cutting parameters.

(3rd paragraph, Section 1) “Currently, research on online monitoring has the following deficiencies. In practical machining, surface roughness varies. The factors affecting surface roughness include machine tool, workpiece, tool characteristics, dynamic parameters, etc., among them, the tool is used to form workpiece surface in machining, which directly affects the surface roughness [24]. For example, as the tool wears, the contact angle between the tool and workpiece changes, which deteriorates the surface roughness of the workpiece [23]. However, the effect of tool wear on surface roughness has been rarely considered in current research [24, 25]. In addition, the deterioration trend of surface roughness and tool wear differs with variable cutting parameters. Consequently, a prediction model trained with the monitoring data collected under one group of cutting parameters can fail to accurately predict roughness when the cutting parameter changes. Evidently, it is unrealistic to train prediction models with all possible cutting parameters. Therefore, to develop a practical surface roughness prediction model, the inclusion of tool wear as one of the variable cutting parameters is necessary. Note that a previous work [26] introduced in detail a method and process for tool condition monitoring. This current study focuses on the surface roughness prediction method. In this paper, the formulation and training of the surface roughness prediction model and its prediction process integrating tool wear are described in detail.”

Point 2: Does the proposed method depend on the type of cutting material and have there been any preliminary attempts to determine the impact of the material?

Author response 2:

Thanks for this valuable comment. The trend of tool wear and surface roughness under different materials may differ, but the process of modeling in this paper can still be applicable, by acquiring machining data from different materials and then adjusting the parameters of the model. In this paper, we do not use many different materials to verify the proposed method for the following reasons. Firstly, the research problem of this paper comes from the machining of a certain type of vertical tail assembly interface, which is made of Ti6Al4V, so we use this material to carry out machining experiments. Secondly, the machining experiment process is costly in terms of complexity and time so we only verified one material at present. But indeed, using the monitoring data of different materials to establish a different roughness prediction model for specific materials is meaningful for aviation manufacturing. We will verify the applicability of the proposed method in different materials in our coming study.

Author action 2:

In Section 4.1, we added the following text to explain the reason why we only verified one material at present.

(2nd paragraph, Section 4.1)” It should be pointed out that the research problem of this paper comes from the machining of a certain type of vertical tail assembly interface, which is made of Ti6Al4V. In addition, the machining experiment process is costly in terms of complexity and time. Therefore, this paper uses material Ti6Al4V to carry out the machining experiments and verify the proposed prediction model.

In the Conclusion and future work part, we added the consideration of different materials in our future work.

(Section 5, Conclusions and future work)”In the future, the means for solving the problem of surface roughness prediction considering different tools and machining materials will be investigated.

Point 3: Has the presented method been verified by numerical methods?

Author response 3:

Thanks for the reviewer’s valuable comments. This article does not verify the prediction results of the method through numerical methods. The reasons are as follows, (1) Numerical method and the data-driven methods (as used in this paper) belong to two different research paths, and the numerical method is out of the scope of this paper. At present, a large number of scholars have studied numerical methods, but we find that numerical methods are difficult to fuse real sensor data into the surface roughness prediction to represent random factors in the machining process, but considering real-time sensor data makes predicting surface roughness more realistic. (2) The effectiveness of the proposed method in this paper has been verified through comparing the predicted surface roughness value with the real surface roughness value measured by a portable instrument, ISR-C300, the error between the predicted value and the real measured value are very small. This verification is more convincing than the numerical methods. Indeed, the numerical method also has the advantages of efficient calculation and strong interpretability. In the future, the combination of the numerical method and the method proposed in this paper is worthy of in-depth study.

Author action 3:

In the second paragraph of section 1, we have added the following text to explain the motivation of surface roughness online prediction using real-time sensor data.

(2nd paragraph, Section 1) “In addition, there are also studies on surface roughness estimation through the numerical method, which is difficult to integrate the real-time sensor data of the machining site to adaptively adjust the roughness prediction online. Given the cons of offline and numerical methods, this paper uses the scheme for surface roughness online prediction based on real-time sensor data.

In the Conclusion and future work, a prospect of combining numerical methods and artificial intelligence models to predict surface roughness has been added.

(Section 5, Conclusions and future work)”Furthermore, the method of predicting surface roughness by a hybrid model that combines the strong interpretable model (numerical method, cutting dynamics, etc.) and artificial intelligence model is worthy of in-depth study.

Reviewer 4 Report

Brief summary

The main goal of the paper was to develop a new model for the prediction of surface texture, especially roughness, by using deep learning and neural networks algorithms. The developed model was supported by a deep literature review. The analysis is well supported graphically. The biggest strength of the paper is an experimental verification of the developed prediction model, proving its usability and applicability for real case scenarios. The authors identified also weak spots of the model and presented a future plan for the model improvements, increasing its reliability and decreasing performance uncertainty.

General concept comments

Article:

I would suggest making small changes to the title. Most of the paper is focusing on the prediction model development, with the application for the surface roughness prediction of the vertical tail assembly. The model can be, in a general view, used for other applications as well. Maybe you can emphasise it somehow, especially since authors use NN and deep learning algorithms for prediction. However, this is only a suggestion.

The weakest part of the paper is that although the developed model is for the prediction of surface roughness, there is little explanation about the roughness itself. Nowadays, one can use more than 200 surface texture parameters, for profile (2D) and surface (3D) assessment. The authors chose the most common one “Ra”. This parameter, however “famous” and known, has a lot of disadvantages. Did the authors consider investigating other surface parameters?

Moreover, there is no information about what type of filtering (for the real case scenario), cut-off, and other parameters to reflect on surface roughness appearance. This information should be added.

I disagree with the statement that the offline surface measurement involves only contact measurement equipment (lines 57-60). There is a quite broad variety of optical measurement equipment which can be used for that purpose. Could you please reflect on this sentence formulation?

It is very unclear how authors calculated tool wear only based on the registered image. What is the quantifier (parameters) used to represent the tool wear? This must be clarified.

All model input parameters should be specified more thoroughly., e.g., by describing parameters units, range of values, etc. This information is really important for all readers, especially practitioners.

The authors presented the vertical tail of aircraft as an example for applying the developed model. However, as the authors mentioned, due to the confidentiality of the manufacturer it was difficult to collect all necessary data of the real aircraft assembly interface. Thus, I’m interested in how authors were able to prepare components, cite “... with high similarity with the real assembly interface ...”? Where does the information about high similarity come from? That should be explained in the text.

Although partly explained, I wish authors explain more clearly why the model requires a small set of data for fine-tuning. The authors identified it as something that needs to be improved, but an explanation is needed.

Do authors mean Acoustic Emission (AE) by writing acoustic? If so, I propose a correct term and abbreviation.

Why X_Ra constitute of a vector with 11 values? Why this number?

Review:

The presented paper is of high importance. The prediction of the surface texture (including roughness) can be helpful in many production occasions. The model development is well presented and motivated. Observations are correct, and recommendations are of high value. It is rather clear, that the authors have deep knowledge of the topic of mathematical models development.

The paper is very well structured. Nevertheless, I recommend using proofreading for language correction and adjustment.

The references are well chosen and are well incorporated into the paper text, supporting statements, and observations.

The developed model has a great potential to be used on the production shop floor for many manufacturing applications related to surface modification. Nevertheless, there is still work that need to be done. To increase the reliability of the model and method the uncertainty budget should be presented. Also, more surface parameters should be investigated.

Specific comments

All tables font size is bigger than regular text. Need to change it.

I recommend changing dark colours on the figure to lighter ones. Text on the dark blue background is not readable. Please, correct it if possible, or at least change the font colour to white.

In several places, I found that authors wrote: “we” (line 416, 512). Please correct sentences and rewrite them in the third person.

There is missing space between the number of references in square brackets, like for example in line 125, or 305.

Line 361: instead of “.. time step..” I would rather use the “timestamp” term.

Figure 11: Consider using different colours for different characteristics. It would improve readability. Now, it looks like the changes of the same parameter, which isn’t.

Table 5: there is an unnecessary coma sign in columns with numbers, e.g, in the column “Output shape” first value “(,5,20)”.

Table 7: in the caption word “Ref.” is not necessary.

Author Response

Response to Reviewer 4 Comments

General concept comments:

Point 1: would suggest making small changes to the title. Most of the paper is focusing on the prediction model development, with the application for the surface roughness prediction of the vertical tail assembly. The model can be, in a general view, used for other applications as well. Maybe you can emphasise it somehow, especially since authors use NN and deep learning algorithms for prediction. However, this is only a suggestion.

Author response 1:

Thanks for the reviewer’s comments. We agree that this method has general characteristics, but its research problem comes from the vertical tail assembly interface. At present, the method proposed in this paper is only verified under the assembly interface material of the vertical tail. In the future, we will use the data of different machining materials and tools to further verify the generalization ability of the proposed method in this paper. Therefore, we think that keeping the title unchanged is consistent with the current research results.

Point 2: The weakest part of the paper is that although the developed model is for the prediction of surface roughness, there is little explanation about the roughness itself. Nowadays, one can use more than 200 surface texture parameters, for profile (2D) and surface (3D) assessment. The authors chose the most common one “Ra”. This parameter, however “famous” and known, has a lot of disadvantages. Did the authors consider investigating other surface parameters?

Author response 2:

Thanks for the reviewer’s comments. We agree with the reviewers that it is indeed a lot of surface texture parameters to evaluate the quality of the machining surface. The reasons for modeling and predicting Ra in this paper are as follows: firstly, the surface roughness Ra is an important parameter concerned by engineers during aviation manufacturing. Secondly, Ra is an important quality evaluation indicator of the vertical tail assembly interface of a certain type of aircraft involved in the experiment. In addition, if the surface quality needs to be comprehensively evaluated, more parameters are needed, such as 3-D surface roughness parameters, surface waviness, surface form, etc. However, it is beyond the research scope of this paper. We will study the prediction of more parameters in the future.

Author action 2:

In Section 1, we added more reviews of surface roughness and explained why we chose surface roughness Ra.

(1st paragraph, Section 1)”There are many parameters affecting surface quality, such as surface roughness (2-D and 3-D surface roughness), surface waviness, surface form, etc. [5, 6], among which, surface roughness parameters and the associated functionality are very important for the evaluation of surface integrity and machining quality [7, 8] since surface roughness significantly influences the assembly accuracy, fatigue strength, corrosion resistance, and contact stiffness of parts [9, 10]. Consequently, surface roughness becomes an important parameter concerned by engineers during aviation manufacturing[11, 12], for instance, in the manufacturing of assembly interface of aircraft vertical, as shown in Figure 1.

In the Conclusion and future work, we added the following text to explain the future research on online prediction of more surface texture parameters.

(Section 5, Conclusions and future work)”There are many other surface texture parameters for evaluating surface quality, such as 3-D surface roughness, surface waviness, surface form, etc. In the future, online monitoring of more parameters should be studied to comprehensively evaluate surface quality.

[5] Aris, N.F.M., Cheng, K. Characterization of the surface functionality on precision machined engineering surfaces. Int. J. Adv. Manuf. Technol. 2008, 38, 402-409.

[6] Grzesik, W. Advanced Machining Processes of Metallic Materials. Elsevier: Amsterdam, 2008; pp. 405-420.

[11] Wang, M.Y., Yang, C.W., Ziliang, L.I., Zhao, S.F., et al. Effects of surface roughness on the aerodynamic performance of a high subsonic compressor airfoil at low Reynolds number. Chin. J. Aeronaut. 2021, 34, 71–81.

[12] Duan, Z.J., Li, C.H., Zhang, Y.B., Dong, L., et al. Milling surface roughness for 7050 aluminum alloy cavity influenced by nozzle position of nanofluid minimum quantity lubrication. Chin. J. Aeronaut. 2021. 34, 33–53.

Point 3: Moreover, there is no information about what type of filtering (for the real case scenario), cut-off, and other parameters to reflect on surface roughness appearance. This information should be added.

Author response 3:

Thanks for the reviewer’s valuable comments. The factors affecting surface roughness include machine tool, workpiece, tool characteristics, dynamic parameters, etc., among them, the tool is used to form workpiece surface in machining, which directly affects the surface roughness. For example, as the tool wears, the contact angle between the tool and workpiece changes, which deteriorates the surface roughness of the workpiece. However, the effect of tool wear on surface roughness has been rarely considered in current research. Therefore, this paper attempts to predict the surface roughness by integrating tool wear and sensor data, so that the surface roughness prediction is more conforming to the actual situation.

In addition, in the machining experiment, the surface roughness is measured after each cutting segment is machined. The surface roughness value is obtained by calculating the average of the surface roughness values of three points in the Z-axis cutting direction of the spindle in each cutting segment, as shown in Figure 10 (c).

Author action 3:

In the third paragraph of Section 1, we added the factors affecting surface roughness in machining to further explain the reason why we chose integrated tool wear.

(3rd paragraph, Section 1)”The factors affecting surface roughness include machine tool, workpiece, tool characteristics, dynamic parameters, etc., among them, the tool is used to form workpiece surface in machining, which directly affects the surface roughness [24].

In addition, in Section4.1, we updated figure 10 and added words to describe our method of collecting surface roughness.

(3rd paragraph, Section 4.1)”The Ra value is obtained by calculating the average of the surface roughness values of three points in the Z-axis cutting direction of the spindle in each cutting segment, as shown in Figure 10 (c).

Point 4: I disagree with the statement that the offline surface measurement involves only contact measurement equipment (lines 57-60). There is a quite broad variety of optical measurement equipment which can be used for that purpose. Could you please reflect on this sentence formulation?

Author response 4:

Thanks to the reviewer's careful comments, the expression of offline surface measurement (lines 57-60) in Section 1 is indeed not rigorous enough, so we added some literature and modified the expression of this sentence.

Author action 4:

We revised the expression of offline surface roughness measurement in the second paragraph of section 1.

(2nd paragraph, Section 1)” Surface roughness is typically monitored either offline or online. The offline approach involves the use of contact measurement equipment [13, 14] and optical measurement equipment [15-17]. The offline approach has drawbacks such as long measurement time, specific requirements to the working environment, complexity to set up, etc.

[14] Dhanasekar, B., Ramamoorthy, B. Restoration of blurred images for surface roughness evaluation using machine vision. Tribol. Int. 2010, 43, 268-276.

[15] Laopornpichayanuwat, W., Visessamit, J., Tianprateep, M. 3-D Surface roughness profile of 316-stainless steel using vertical scanning interferometry with a superluminescent diode. Measurement. 2012, 45, 2400-2406.

[16] Nwaogu, U.C., Tiedje, N. S., Hansen, H. N. A non-contact 3D method to characterize the surface roughness of castings. J. Mater. Process. Technol. 2013, 213, 59-68.

[17] SamtaÅŸ, G. Measurement and evaluation of surface roughness based on optic system using image processing and artificial neural network. Int. J. Adv. Manuf. Technol. 2014, 73, 353–364.

Point 5: It is very unclear how authors calculated tool wear only based on the registered image. What is the quantifier (parameters) used to represent the tool wear? This must be clarified.

Author response 5:

Thanks for this valuable comment. The tool wear value is measured by the measurement software provided by the portable microscope, Dino-Lite AM7115MZT. After each cutting segment is machined, a picture of tool wear is taken and then magnified. Then, the value of tool wear can be obtained by using the ruler in the measurement software.

Author action 5:

We added a screenshot of the measurement software in Figure 10 (b) to show how the tool wear is measured and added a sentence to describe the measurement process.

(3rd paragraph, Section 4.1)”The VB value of the cutting tool's flank wear is measured by the ruler of the measurement software provided by the portable microscope, Dino-Lite AM7115MZT as shown in Figure 10 (b).

Point 6: All model input parameters should be specified more thoroughly., e.g., by describing parameters units, range of values, etc. This information is really important for all readers, especially practitioners.

Author response 6:

Thanks for this valuable comment. The unit of tool wear (VB) and surface roughness (Ra) is microns (μm), the unit of the sound pressure signal is Pascal (Pa), the unit of vibration signal is gravity acceleration 9.8m/s2 (g), and the unit of the current signal is Ampere (A). The values of VB and Ra are actual measured or predicted values, and the measured values of the sensor signal are given in Figure 11.

Author action 6:

We added the following text in section 3.4.1 to describe the parameters of the model input data

(1st paragraph, Section 3.4.1)”The unit of tool wear (VB) and surface roughness (Ra) is microns (μm), the unit of the sound pressure signal is Pascal (Pa), the unit of vibration signal is gravity acceleration 9.8m/s2 (g), and the unit of the current signal is Ampere (A). The values of VB and Ra are actual measured or predicted values, and the measured values of the sensor signal are given in Figure 11.

Point 7: The authors presented the vertical tail of aircraft as an example for applying the developed model. However, as the authors mentioned, due to the confidentiality of the manufacturer it was difficult to collect all necessary data of the real aircraft assembly interface. Thus, I’m interested in how authors were able to prepare components, cite “... with high similarity with the real assembly interface ...”? Where does the information about high similarity come from? That should be explained in the text.

Author response 7:

Thanks for the reviewer’s comments. The manufacturer is only strict in the management of the machining process and machining process data, i.e., it is difficult to collect sensor data during the machining process for experimental analysis. But, we can obtain the material of assembly interface, machining parameter range, tool type, surface quality requirements, geometric dimensions, and other information of the vertical tail assembly interface. This information is enough for us to design and fabricated a high similar sample workpiece of the assembly interface, simulate the machining process of the real assembly interface in the experimental environment, and design a complex experimental data acquisition scheme. Therefore, the experiment designed in this paper can be highly similar to the vertical tail assembly interface in machining conditions.

Author action 7:

We updated some contents in Section 4.1 to explain that the sample workpiece of the sub-assembly interface designed in this paper and its machining process is similar to the real assembly interface.

(1st paragraph, Section 4.1) “Due to the strict management of the machining process of the aircraft assembly interface, it is difficult to collect sensor data during the machining process of the real assembly interface. But the material of assembly interface, machining parameter range, tool type, surface quality requirements, geometric dimensions, and other information of the vertical tail assembly interface are available. This information is enough for us to design and fabricated a high similar sample workpiece of the assembly interface, simulate the machining process of the real assembly interface in the experimental environment, and design a complex experimental data acquisition scheme. Therefore, the milling experiment of the sample workpiece is adequately capable of simulating the real assembly interface, and the proposed method can be validated by collecting the data during machining.

Point 8: Although partly explained, I wish authors explain more clearly why the model requires a small set of data for fine-tuning. The authors identified it as something that needs to be improved, but an explanation is needed.

Author response 8:

Thanks for this valuable comment. We need to explain that the transfer learning strategy proposed in this paper includes two stages: pre-training and fine-tuning. In the pre-training stage, the SAE-LSTM model is pre-trained with source domain data (source domain model), so that it can predict surface roughness well in the source domain. In the fine-tuning stage, fine-tuning feature extraction module and prediction module based on source domain model are required. It should be pointed out that in the first step of the fine-tuning stage, based on the trained model weight of the source domain model, we use the unlabeled target domain data to fine-tune the feature extraction module of the prediction model. MMD is used to reduce the difference between source domain data and target domain data so that the source domain model can adapt to the target domain. However, due to the huge difference between the source domain and the target domain, only using the unlabeled target domain data to train the feature extraction module can not obtain a high-precision prediction effect. The strategy of using a small number of target domain label data to adjust the prediction module has been proved to improve the efficiency and accuracy of model training in the transfer learning method [53], and a small amount of label data in the target domain is usually easy to obtain. Therefore, we freeze the weight of the feature extraction module that has been fine-tuned, so that the prediction module is only participated in fine-tuning. In this way, it can obtain high prediction accuracy without retraining the model with a large number of label data in the target domain.

Author action 8:

In section 3.3.1, we added the following text to explain why the strategy of transfer learning in this paper and the need for a small set of data to fine-tune the model.

(3rd paragraph, Section 3.3.1)”In the fine-tuning stage, this paper needs to fine-tune the feature extraction module and prediction module based on the source domain model. In the third stage, a double-input structure model based on the source domain model is designed. The source and target domain data are used as inputs of the double-input model. The feature extraction module extracts feature from the two domains. It should be noted that the target domain data here does not need label data. The distribution difference of features extracted from the two domains is computed in a customized layer, which quantifies the distribution difference between the source and target domain data using the maximum mean discrepancy (MMD). Then, the error between the distribution difference and zero is back-propagated to optimize the parameters of the SAE–LSTM model. Accordingly, the source domain model can adapt to the target domain. However, due to the huge difference between the source domain and the target domain, only using the unlabeled target domain data to train the feature extraction module can not obtain a high-precision prediction effect. The strategy of using a small number of target domain label data to adjust the prediction module has been proved to improve the efficiency and accuracy of model training in the transfer learning method [53], and a small amount of label data in the target domain is usually easy to obtain. Therefore, in the fourth stage, the model is adjusted to a single input, and the weight of the fine-tuned feature extraction module is frozen so that the prediction module only participates in the fine-tuning. In this way, it can obtain high prediction accuracy without retraining the model with a large number of label data in the target domain.

[53]Zheng, J.M., Cai, F., Chen, H.H., Rijke, M.D. Pre-train, Interact, Fine-tune: a novel interaction representation for text classification. Inf. Process. Manage. 2020, 57, 102215.

Point 9: Do authors mean Acoustic Emission (AE) by writing acoustic? If so, I propose a correct term and abbreviation.

Author response 9:

Thanks for the reviewer's comments. The sensor used in the experiment designed in this paper is a sound pressure sensor, not an acoustic emission sensor. Therefore, we modified the expression of acoustic in the article to avoid misunderstanding.

Author action 9:

We modified the word “acoustic” in the article to “sound pressure”.

Point 10: Why X_Ra constitute of a vector with 11 values? Why this number?

Author response 10:

Thanks for the reviewer's comments. In this paper, surface roughness prediction integrates tool wear and sensor data. Based on our prior knowledge, the following signals, cutting force, vibration, noise, and spindle load, are sensitive to tool wear (i.e., the amplitude of the signals increases with tool wear) and thus are good indicators for tool wear. On the other hand, vibration, noise, and current signals are cheap to collect due to that the sensors are economical and convenient to install. Therefore, we choose these physical quantities to develop the surface roughness prediction model. It should be noted that cutting force is most directly related to tool wear and surface roughness and is used by many researchers. However, cutting force sensor has many restrictions and is impractical in real applications, such as very inconvenient installation for the large workpiece and high cost, and thus we abandoned the cutting force signal acquisition. Therefore, this paper uses three kinds of sensors: sound pressure, vibration, and current. The vibration of the spindle can directly reflect the vibration of the tool. In order to reflect the influence of the spindle vibration, we have installed a three-axis acceleration vibration sensor on the spindle and spindle box, respectively. The three-phase (U, V, C) current signal is measured on the spindle controller, and the sound pressure sensor is used to measure the sound in machining. Finally, surface roughness input data X_ Ra contains 1 tool wear data vector, 6 vibration signal data vectors, 3 current signal data vectors, and 1 sound pressure signal data vector, a total of 11.

Author action 10:

We have added the following text in section 4.1 to explain the reason for use of vibration sensors, current sensors, and sound pressure sensors. 

(3rd paragraph, Section 4.1)” Based on the prior knowledge, the following signals, vibration, noise, and spindle load, are sensitive to tool wear (i.e., the amplitude of the signals increases with tool wear) and thus are good indicators for tool wear and further for surface roughness. On the other hand, vibration, noise, and current signals are cheap to collect since the sensors are economical and convenient to install. Therefore, this paper chooses these physical quantities to develop the surface roughness prediction model. It should be noted that cutting force is most directly related to tool wear and surface roughness and is used by many researchers. However, cutting force sensor has many restrictions and is impractical in real applications, such as very inconvenient installation for the large workpiece and high cost, and thus this paper abandoned the cutting force signal acquisition.

Review:

The presented paper is of high importance. The prediction of the surface texture (including roughness) can be helpful in many production occasions. The model development is well presented and motivated. Observations are correct, and recommendations are of high value. It is rather clear, that the authors have deep knowledge of the topic of mathematical models development.

The paper is very well structured. Nevertheless, I recommend using proofreading for language correction and adjustment.

The references are well chosen and are well incorporated into the paper text, supporting statements, and observations.

The developed model has a great potential to be used on the production shop floor for many manufacturing applications related to surface modification. Nevertheless, there is still work that need to be done. To increase the reliability of the model and method the uncertainty budget should be presented. Also, more surface parameters should be investigated.

Author response 11:

Thanks for the reviewer’s positive affirmation of this paper. We have made a point-to-point response to the general concept comments above, and we have proofread the entire manuscript to eliminate the errors. As the replies of point10 and point6 in the general concept comments, we explained the budget of the experimental setting in this paper and more research on surface roughness parameters. Thanks very much again for your help.

Specific comments:

All tables font size is bigger than regular text. Need to change it.

I recommend changing dark colours on the figure to lighter ones. Text on the dark blue background is not readable. Please, correct it if possible, or at least change the font colour to white.

In several places, I found that authors wrote: “we” (line 416, 512). Please correct sentences and rewrite them in the third person.

There is missing space between the number of references in square brackets, like for example in line 125, or 305.

Line 361: instead of “.. time step..” I would rather use the “timestamp” term.

Figure 11: Consider using different colours for different characteristics. It would improve readability. Now, it looks like the changes of the same parameter, which isn’t.

Table 5: there is an unnecessary coma sign in columns with numbers, e.g, in the column “Output shape” first value “(,5,20)”.

Table 7: in the caption word “Ref.” is not necessary.

Author response 12:

Thanks for the reviewer’s careful reading and specific comments. We have made corresponding modifications in the article for all the above specific comments. And we have proofread the entire manuscript to eliminate the errors. Thanks very much again for your help.
